# CBD: A Certified Backdoor Detector Based on Local Dominant Probability

**Zhen Xiang**      **Zidi Xiong**      **Bo Li**
University of Illinois Urbana-Champaign
{zxiangaa, zidix2, lbo}@illinois.edu

## Abstract

Backdoor attack is a common threat to deep neural networks. During testing, samples embedded with a backdoor trigger will be misclassified as an adversarial target by a backdoored model, while samples without the backdoor trigger will be correctly classified. In this paper, we present the first *certified backdoor detector* (CBD), which is based on a novel, adjustable conformal prediction scheme based on our proposed statistic *local dominant probability*. For any classifier under inspection, CBD provides 1) a detection inference, 2) the condition under which the attacks are guaranteed to be detectable for the same classification domain, and 3) a probabilistic upper bound for the false positive rate. Our theoretical results show that attacks with triggers that are more resilient to test-time noise and have smaller perturbation magnitudes are more likely to be detected with guarantees. Moreover, we conduct extensive experiments on four benchmark datasets considering various backdoor types, such as BadNet, CB, and Blend. CBD achieves comparable or even higher detection accuracy than state-of-the-art detectors, and it in addition provides detection certification. Notably, for backdoor attacks with random perturbation triggers bounded by $\ell_2 \leq 0.75$ which achieves more than 90% attack success rate, CBD achieves 100% (98%), 100% (84%), 98% (98%), and 72% (40%) empirical (certified) detection true positive rates on the four benchmark datasets GTSRB, SVHN, CIFAR-10, and TinyImageNet, respectively, with low false positive rates.

## 1 Introduction

Despite the success of deep neural networks (DNNs) in many applications, they are vulnerable to adversarial attacks such as backdoor attacks [46, 36]. A DNN being backdoored will learn to predict an adversarial target class for test samples embedded with a backdoor trigger, while maintaining a high accuracy on clean test samples without the trigger [18, 8, 42, 65, 50, 43, 51, 37].

Backdoor detection is a popular task for backdoor defense. It aims to detect if a given model is backdoored without access to the training set or any real test samples that are possibly embedded with the trigger [48, 64]. The task corresponds to practical scenarios where the user of an app or a legacy system containing a DNN seeks to know if the model is backdoor attacked, where the training set is not available [1]. Various empirical approaches have been proposed for backdoor detection, most of which are based on trigger reverse engineering [67, 6, 75, 69, 56, 62, 24], or meta classification [82, 29]. However, none of these works quantitatively investigate the conditions under which the backdoor attacks are guaranteed to be detectable. Without a detection guarantee, DNNs are still vulnerable to future attacks (e.g.) with new trigger types [51, 87, 70].

In this paper, we make the first attempt toward the *certification* of backdoor detection. Certification is an important concept for studies on the robustness of DNNs against adversarial examples [61, 17, 52, 7, 45]. In particular, a robustness certification refers to a probabilistic or deterministic guarantee for a model to produce desired outputs (e.g. correct label prediction) when the adversarial perturbation applied to the inputs satisfies certain conditions (e.g. with some constrained perturbation magnitude) [33, 58, 32, 10, 59]. As an analogy, we propose a *certified backdoor detector* (CBD)

37th Conference on Neural Information Processing Systems (NeurIPS 2023).

that is guaranteed to trigger an alarm if the attack for a given domain satisfies certain conditions. In Sec. 3, we introduce the detection procedure of CBD based on conformal prediction (with our proposed, optional adjustment scheme) using a novel (model-level) statistic named *local dominant probability*. The calibration set for conformal prediction is obtained from a small number of benign shadow models trained on a small validation set, which addresses the unavailability of the training set. In Sec. 4, we derive the condition for attacks with detection guarantee, as well as a probabilistic upper bound for the false positive rate of our detector, for any prescribed significance level specifying the aggressiveness of the detection. Notably, our certification is comprehensive – for any domain, more effective attacks with strong *trigger robustness* (which measures the resilience of a trigger against test-time noises) and more stealthy attacks (against human inspectors) with small trigger perturbation magnitudes are easier to be detected with guarantees. Moreover, both our detector and the certification method do not assume the trigger incorporation mechanism or the training configuration of the model, which allows their potential application to future attacks. Our contributions are summarized below:

- We propose CBD, the first certified backdoor detector, which is based on an adjustable conformal prediction scheme using a novel *local dominant probability* statistic.
- We propose a certification method and show that for any domain, backdoor attacks with stronger *trigger robustness* and smaller trigger perturbation magnitudes are more likely to be detected by CBD with guarantee. We also derive a probabilistic upper bound for the false positive rate of CBD.
- We show that CBD achieves comparable or even higher detection accuracy than state-of-the-art detectors against three types of backdoors. We also show that for backdoor attacks with random perturbation triggers bounded by $\ell_2 \leq 0.75$, CBD achieves 100% (98%), 100% (84%), 98% (98%), and 72% (40%) empirical (certified) true positive rates on GTSRB, SVHN, CIFAR-10, and TinyImageNet, respectively, with only 0%, 0%, 6%, and 10% false positive rates, respectively.

## 2 Related Work

**Backdoor Detection Methods** Existing methods for backdoor detection are all empirical without theoretical guarantees. An important type of reverse-engineering-based method estimates putative triggers for anomaly detection [67, 6, 75, 41, 78, 73, 79, 76, 69, 56, 62, 24]. Certification for these methods is hard due to the complexity of trigger reverse engineering. Another type of detector is based on meta-classification that involves a large number of shadow models trained with and without attacks [82, 29]. Differently, our CBD is based on conformal prediction involving a scalar detection statistic (rather than a high dimensional feature vector employed by [82, 29]), such that only a small number of benign shadow models will be required. More importantly, our CBD is certified, i.e. with a detection guarantee, which is different from all the detention methods mentioned above.

**Other Backdoor Defense Tasks** Backdoor defenses during training aim to produce a backdoor-free classifier from the possibly poisoned training set [63, 5, 74, 14, 25, 57, 77, 23, 4, 16]. These defenses require access to the training set, which is unavailable for backdoor detection. Backdoor mitigation aims to "repair" models being backdoor attacked [40, 72, 35, 19, 88, 85], which can be viewed as a downstream task following backdoor detection. Inference-stage trigger detection aims to detect if a test sample contains the trigger [15, 55, 12, 9, 44, 34]. However, backdoor detection is performed before the inference stage, where test samples are not available. These defense tasks will not be further discussed due to their fundamental differences from the backdoor detection task.

**Certified Backdoor Defenses** Existing methods mostly modify the training process to prevent the backdoor from being learned, while manipulating the test sample to destroy any potentially embedded triggers [86, 71, 27, 53]. These methods are not applicable to the backdoor detection problem where both the training set and the test samples are not available. More importantly, all these existing certified defenses are deployed during training, which requires an uncontaminated training process fully controlled by the defender. In contrast, in this paper, we consider a stronger threat model that allows the attacker to have full control of the training process.

## 3 Detection Method

### 3.1 Problem Definition

**Threat Model** Consider a classification domain with sample space $\mathcal{X}$ and label space $\mathcal{Y}$. A backdoor attack is specified by a trigger with some incorporation function $\delta : \mathcal{X} \to \mathcal{X}$ and a target class $t \in \mathcal{Y}$. For a successful backdoor attack, the victim classifier will predict to the target class $t$ whenever a

test sample is embedded with the trigger, while test samples without the trigger will be correctly classified [18, 8, 42, 38]. In this paper, we do allow advanced attackers with full control of the training process [51, 87, 70, 50, 3]. This is deemed a stronger threat model than all previous works on certified backdoor defense where the defender is assumed with full control of the training process. However, in this paper, we do not consider backdoor attacks with multiple triggers or target classes [84, 83] – even empirical detection of these attacks is a challenging problem [80].

**Goal of Certified Backdoor Detection** The fundamental goal is backdoor detection, i.e. to infer if a given classifier $f(\cdot; w) : \mathcal{X} \to \mathcal{Y}$ is backdoored or not [48, 64]. The defender has no access to the training set or any test samples that may contain the trigger. In practice, the defender also has no access to benign classifiers with high accuracy for the same domain as $f(\cdot; w)$ – otherwise, these benign classifiers can be used for the task, and detection will be unnecessary. However, the defender is assumed with a small validation set of clean samples for detection – this is a standard assumption made by most post-training backdoor defenses [67, 82, 75, 56, 41].

Beyond model inference, a *certified* backdoor detector also needs to provide each classification domain (associated with the model to be inspected) with a condition on $\delta$, $t$, and $w$, such that any successful backdoor attack with a trigger $\delta$ and a target class $t$ on a victim classifier $f(\cdot; w)$ (trained on this classification domain) is guaranteed to be detected if the condition is satisfied. The stronger a certification is, the more likely a successful attack on the domain will be detectable with guarantee. Moreover, a certification has to be associated with a guarantee on the false positive rate; otherwise, an arbitrarily strong certification can be achieved by increasing the aggressiveness of the detection rule.

### 3.2 Overview of CBD Detection

**Key Intuition** For a successful backdoor attack with a trigger $\delta$ and a target class $t$, a test instance $x$ embedded with the trigger (denoted by $\delta(x)$) will be classified to the target class $t$ with high probability. Practical backdoor triggers should also be robust (i.e. resilient) to noises either from the environment or introduced by simple defenses based on input-preprocessing such as blurring and/or quantization [81, 39]. Such *trigger robustness* can be measured by the distribution of the model prediction in the neighborhood of each $\delta(x)$ – the more robust the trigger is, the more samples in the neighborhood of $\delta(x)$ will be predicted to the target class $t$. Thus, if the perturbation magnitude of a robust trigger, i.e. $||\delta(x) - x||_2$, is small (which is usual for backdoor attacks to achieve stealthiness), a significant proportion of samples in the neighborhood of $x$ will also be classified to class $t$ due to their closeness to $\delta(x)$. Such an increment in the target class probability in the neighborhood of all instances is captured by our proposed statistic named *local dominant probability* (LDP) to distinguish backdoored classifiers from benign ones – the former tend to have a larger LDP than the latter.

**Outline of CBD Detection Procedure** In short, CBD performs an *adjustable* conformal prediction to test if the LDP statistic computed for the model to be inspected is sufficiently large to trigger an alarm. A small number of shadow models (with the same architecture as the model to be inspected) are trained on a relatively small validation set, with an LDP computed for each shadow model to form a calibration set. Then, a small proportion of outliers with large values in the calibration set are optionally removed, e.g., by anomaly detection. Based on this adjusted calibration set, the p-value is computed for the LDP for the model to be inspected. The model is deemed to be backdoored if the p-value is less than some prescribed significance level (e.g. the classical 0.05 for statistical testing).

### 3.3 Definition of LDP

We first define a *samplewise local probability vector* (SLPV) to represent the distribution of the model prediction outcomes in the neighborhood of any given sample. Then, based on SLPV, we define the *samplewise trigger robustness* (STR) that measures the resilience of a trigger to the Gaussian noise when it is embedded in a particular sample. Finally, we define the LDP statistic based on SLPV.

**Definition 3.1.** *(Samplewise Local Probability Vector (SLPV)) For any classifier $f(\cdot; w) : \mathcal{X} \to \mathcal{Y}$ with parameters $w$, the SLPV for any input $x \in \mathcal{X}$ is a probability vector $\boldsymbol{p}(x|w, \sigma) \in [0, 1]^K$ over the $K = |\mathcal{Y}|$ classes, with the $k$-th entry defined by $p_k(x|w, \sigma) \triangleq \mathbb{P}_{\epsilon \sim \mathcal{N}(0, \sigma^2 I)}(f(x + \epsilon; w) = k)$, $\forall k \in \mathcal{Y}$, where $\mathcal{N}(\mu, \Sigma)$ represents Gaussian distribution with mean $\mu$ and covariance $\Sigma$.*

**Remarks:** If $f(\cdot; w)$ is continuous at $x$ with $f(x; w) = k$ for some $k \in \mathcal{Y}$, it is trivial to show that $p_k(x|w, \sigma) \to 1$ as $\sigma \to 0$, i.e. the SLPV becomes a singleton at the predicted class of $x$ without the Gaussian noise. In practice, SLPV can be estimated by Monte Carlo for any given model and sample.

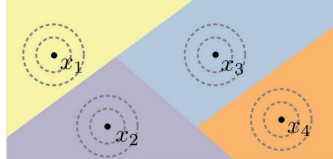 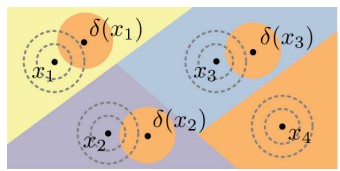

(a) Benign classifier with a small LDP close to $\frac{1}{4}$.  (b) Classifier being backdoored with a large LDP.

Figure 1: Illustration of the difference in LDP between benign and backdoored classifiers on a classification domain with $K = 4$ classes. A backdoor attack with a robust trigger $\delta$ (with a small perturbation magnitude) and a target class 4 (with orange decision region) changes the class distribution in the neighborhood of $x_1$, $x_2$, and $x_3$, resulting in a larger LDP for the backdoored model than for the benign model.

**Definition 3.2.** *(Samplewise Trigger Robustness (STR)) Consider a backdoor attack with a trigger $\delta$ and a target class $t \in \mathcal{Y}$ against a victim model $f(\cdot; w)$. For any sample $x \in \mathcal{X}$ and any isotropic Gaussian distribution $\mathcal{N}(0, \sigma^2 I)$, the STR is defined by the $t$-th entry of the SLPV for $\delta(x)$ (i.e. sample $x$ with the trigger $\delta$ embedded), which is denoted by $R_{\delta,t}(x|w, \sigma) \triangleq p_t(\delta(x)|w, \sigma)$.*

> **Remarks:** STR measures the resilience of a trigger against Gaussian noises. Usually, strong STR can be naturally achieved by embedding the trigger in a large variety of samples during training.

**Definition 3.3.** *(Local Dominant Probability (LDP)) Consider a domain with $K = |\mathcal{Y}|$ classes and an isotropic Gaussian distribution $\mathcal{N}(0, \sigma^2 I)$. The LDP for a classifier $f(\cdot; w)$ is defined by*

$$s(w) = \|\frac{1}{K}\sum_{k=1}^{K} \boldsymbol{p}(x_k|w, \sigma)\|_\infty \tag{1}$$

*where $x_1, \cdots, x_K$ are $K$ independent random samples satisfying $f(X_k; w) = k, \forall k \in \{1, \cdots, K\}$.*

> **Remarks:** By the definition, LDP for a given classifier $f(\cdot; w)$ is computed on $x_1, \cdots, x_K$ independently sampled from the $K$ classes respectively, satisfying $f(x_k; w) = k$ for $\forall k$. LDP computed on more samples per class yields similar detection and certification performance empirically. Specifically, we first compute the SLPV for each of the $K$ samples. Then we take the average of the $K$ SLPVs and pick the maximum entry as the LDP for $f(\cdot; w)$. Note that LDP is always no less than $\frac{1}{K}$.

As stated in the key intuitions in Sec. 3.2, LDP tends to be larger for backdoored classifiers than for benign ones. This can be understood from the illustration in Fig. 1 for a classification domain with $K = 4$ classes. For the benign classifier on the left, the SLPVs for $x_1, \cdots, x_4$ are almost orthogonal to each other, leading to a small LDP close to $\frac{1}{4}$. On the right, we consider a backdoor attack with a robust trigger $\delta$ and a target class 4 (with orange decision region). The strong STRs for $x_1$, $x_2$, and $x_3$ (represented by the large orange regions around $\delta(x_1)$, $\delta(x_2)$, and $\delta(x_3)$), together with the small trigger perturbation magnitudes (i.e. small $\|\delta(x_i) - x_i\|_2$ for $i = 1, 2, 3$), significantly change the class distribution in the neighborhood of each of $x_1$, $x_2$, and $x_3$. In particular, there will be a clear increment in the 4-th entry of the SLPVs for $x_1$, $x_2$, and $x_3$. Thus, the 4-th entry associated with the backdoor target class will dominate the average SLPV over $x_1, \cdots, x_4$, leading to a large LDP.

### 3.4 CBD Detection Procedure

Although backdoored and benign classifiers have different LDP distributions, it is still a challenge in practice to set a detection threshold. To solve this problem, we propose to use the conformal prediction, which employs a calibration set for supervision [66, 2]. Here, the calibration set consists of LDP statistics obtained from a small number of benign shadow models trained on the small validation set possessed by the defender. However, the actual benign classifiers to be inspected are usually trained on more abundant data, such that the LDPs for these classifiers will likely follow a different distribution from the LDPs for the shadow models without sufficient training. In particular, the LDPs in the calibration set (obtained from the shadow models) may easily have an overly large sample variance and a heavy tail of large outliers. Directly using this calibration set for conformal prediction may lead to a conservative detection due to an overly large detection threshold.

Thus, we propose an optional adjustment scheme that treats the $m$ largest LDP statistics in the calibration set as outliers. In practice, prior knowledge may be required to determine the exact value

of $m$, while in our experiments, a small $m/N$ ($\leq 0.2$, where $N$ is the size of the calibration set) may significantly increase the detection or certification power of our CBD with only small increment to the false positive rate. The detection procedure of our CBD consists of the following four steps:

**1)** Given a classifier $f(\cdot; w)$ to be inspected, estimate LDP $s(w)$ based on Def. 3.3.

**2)** Train (benign) shadow models $f(\cdot; w_1), \cdots, f(\cdot; w_N)$ on the clean validation dataset, and construct a calibration set $\mathcal{S}_N = \{s(w_1), \cdots, s(w_N)\}$ by computing the LDP for each model.

**3)** Compute the adjusted conformal p-value (assuming $m$ large outliers) defined by:

$$q_m(w) = 1 - \frac{1 + \min\{|\{s \in \mathcal{S}_N : s < s(w)\}|, N - m\}}{N - m + 1} \tag{2}$$

**4)** Trigger an alarm if $q_m(w) \leq \alpha$, where $\alpha$ is a prescribed significance level (e.g. $\alpha$=0.05).

## 4 CBD Certification

In addition to a detection inference, CBD also provides a certification, which is a condition under which attacks are guaranteed to be detectable. Detailed proofs in this section are shown in App. A.

**Theorem 4.1.** *(**Backdoor Detection Guarantee**) For an arbitrary classifier $f(\cdot; w) : \mathcal{X} \to \mathcal{Y}$ to be inspected, let $x_1, \cdots, x_K$ be the $K$ randomly selected samples and $\mathcal{N}(0, \sigma^2 I)$ be the isotropic Gaussian distribution used to compute the LDP for $f(\cdot; w)$. Let $\alpha$ be the prescribed significance level of CBD. A backdoor attack with a trigger $\delta$ and a target class $t$ is guaranteed to be detected if:*

$$\Delta < \sigma(\Phi^{-1}(1 - s_{(N-m-\lfloor \alpha(N-m+1)\rfloor)}) - \Phi^{-1}(1 - \pi)) \tag{3}$$

*where $\Phi$ is the standard Gaussian CDF, $\pi = \min_{k=1,\cdots,K} R_{\delta,t}(x_k|w, \sigma)$ is the minimum STR over $x_1, \cdots, x_K$, $\Delta = \max_{k=1,\cdots,K} \|\delta(x_k) - x_k\|_2$ is the maximum perturbation magnitude of the trigger, $m$ is the number of assumed outliers in the calibration set $\mathcal{S}_N$ with size $N$, and $s_{(n)}$ denotes the $n$-th smallest element in $\mathcal{S}_N$.*

*Proof (sketch).* The STR for each sample $x_k$ equals the $t$-th entry of the SLPV for $\delta(x_k)$ by Def. 3.2. We also connect the SLPV for $\delta(x_k)$ to the SLPV for $x_k$ using the Neyman-Pearson lemma ([49]). Based on this connection, we derive the lower bound for the minimum STR, such that the $t$-th entry of the SLPV of each $x_k$ is sufficiently large to result in a large LDP for the attack to be detected. □

> **Remarks:** (1) (**Main Results**) For fixed trigger perturbation size ($\Delta$), detection of attacks with larger STR ($\pi$) is more likely to be guaranteed; while for fixed STR, detection of attacks with smaller trigger perturbation size is more likely to be guaranteed. (2) Our backdoor detection guarantee is inspired by the randomized smoothing approach in [10] for certified robustness against adversarial examples. However, certified backdoor detection and certified robustness against adversarial examples are fundamentally different, as will be detailed in App. I. (3) Certified backdoor detection and certified robustness against backdoors complement each other. The former provides detection guarantees to strong backdoor attacks, while the latter prevents the trigger from being learned during training [86, 53]. The two types of certification may cover the entire attack space together in the future, such that a backdoor attack will be either strong enough to be detected or weak enough to be disabled.

A meaningful certification for backdoor detection should also be along with a guarantee for the false positive rate (FPR). Otherwise, one can easily design a backdoor detector that always triggers an alarm, which provides detection guarantees to all backdoor attacks but is useless in practice.

**Theorem 4.2.** *(**Probabilistic Upper Bound for FPR**) Consider a random calibration set $\mathcal{S}_N = \{s_1, \cdots, s_N\}$ with $s_1, \cdots, s_N$ i.i.d. following some continuous distribution $F$. Consider a random benign classifier $f(\cdot; W)$ with LDP $s(W)$ following some distribution $\tilde{F}$. Assume $F$ dominates $\tilde{F}$ in the sense of first-order stochastic dominance. Let $m$ be any number of assumed outliers in $\mathcal{S}_N$ and let $\alpha$ be any prescribed significance level of CBD. Then, the FPR of CBD on $f(\cdot; W)$ conditioned on $\mathcal{S}_N$, which is denoted by $Z_N = \mathbb{P}(q_m(W) \leq \alpha | \mathcal{S}_N)$ based on Eq. (2), will be first-order dominated by a random variable $B$ following $\text{Beta}(m + l + 1, N - m - l)$ with $l = \lfloor \alpha(N - m + 1) \rfloor$, i.e. $B \succeq_1 Z_N$. In other words, $\mathbb{P}(Z_N \leq q) \geq \mathbb{P}(B \leq q)$ for any real $q$.*

*Proof (sketch).* We first express the false positive rate $Z_N$ in terms of the order statistics on the elements of the random calibration set $\mathcal{S}_N$. Then, we derive the lower bound of the CDF of $Z_N$ using the distribution of order statistics followed by a binomial expansion. □

**Remarks:** The assumption that $F$ dominates $\tilde{F}$ in Thm. 4.2 generally holds in practice. Again, this is because the actual benign classifiers to be inspected are usually trained on more abundant data than the benign shadow models. Empirical results supporting this assumption are shown in Sec. 5.4. Moreover, an analysis of this phenomenon is presented in App. B, where we show on binary Bayes classifiers that a higher empirical loss can easily lead to a larger expected LDP.

While we have shown that $Z_N$ conditioned on a random calibration set of size $N$ is upper bounded by a Beta random variable in the sense of first-order stochastic dominance, in Col. 4.3 below, we show that asymptotically, the upper bound of $Z_N$ converges to a constant in probability as $N \to \infty$.

**Corollary 4.3.** *(Asymptotic Property of FPR) Consider the settings in Thm. 4.2. For any $\xi > 0$,* $\lim_{N \to +\infty} \mathbb{P}(Z_N \leq \tau) = 1$, *where $\tau = \alpha + (1-\alpha)\beta + \xi$ with $\beta = m/N$.*

*Proof (sketch).* We show that any random variable $B$ with the Beta distribution described in Thm. 4.2 satisfies $\lim_{N \to +\infty} \mathbb{P}(B \leq \tau) = 1$. Then, the corollary is proved since $Z_N$ is dominated by $B$. $\square$

**Remarks:** For classical conformal prediction without adjustment where $m = 0$, we will have $\beta = 0$ and $\lim_{N \to +\infty} \mathbb{P}(Z_N \leq \alpha + \xi) = 1$ for any $\xi > 0$. In this case, the upper bound of the false positive rate of our CBD converges to the prescribed significance level $\alpha$ in probability.

## 5 Experiment

There have been many different types of backdoor attacks proposed, each also with a wide range of configurations. Thus, it is infeasible to evaluate our CBD over the entire space of backdoor attacks. Inspired by the evaluation protocol for certified robustness [10], in Sec. 5.1, we focus on backdoor attacks with random perturbation triggers to comprehensively evaluate the certification capability of our CBD. In Sec. 5.2, we compare CBD with three state-of-the-art backdoor detectors (all "uncertified") against backdoor attacks with three popular trigger types to evaluate the detection capability of CBD. In Sec. 5.3, we present ablation studies (e.g. on the number of shadow models used by CBD). Additional results and other supportive empirical analyses are shown in Sec. 5.4.

### 5.1 Evaluation of CBD Certification

For certified robustness, the prediction of a test example is unchanged with a guarantee, if the magnitude of the adversarial perturbation is smaller than the *certified radius* [10]. Certified robustness is usually evaluated by the *certified accuracy* on some random test set as the proportion of the samples that are guaranteed to be correctly classified if the magnitude of the adversarial perturbation is no larger than some prescribed value. As an analog, our certification for backdoor detection is specified by an inequality that involves both the STR and the perturbation magnitude of the trigger, which naturally produces a two-dimensional "*certified region*" (illustrated in Fig. 7 and Fig. 8 in Apdx. D). Our certification method is evaluated on a set of random backdoor attacks, each using a random pattern as the trigger, with the perturbation magnitude satisfying some $\ell_2$ constraint. We are interested in the proportion of the attacks falling into the certified region, i.e. guaranteed to be detected.

#### 5.1.1 Setup

**Dataset:** Our experiments are conducted on four benchmark image datasets, GTSRB [60], SVHN [47], CIFAR-10 [30], and TinyImageNet [11], following their standard train-test splits. Due to the large number of models that will be trained to evaluate our certification method, except for GTSRB, we use 40% of the training set to train these models. We also reserve 5,000 samples from the test set of GTSRB, SVHN, and CIFAR-10, and 10,000 samples from the test set of TinyImageNet (much smaller than the training size for the models for evaluation) for the defender to train the shadow models. More details about these datasets are deferred to App. C.1.

**Evaluation Metric:** Following the convention, the detection performance of CBD is evaluated by the true positive rate (**TPR**) defined by the proportion of attacks being detected (with a correct inference of the target class) and the false positive rate (**FPR**) defined by the proportion of benign classifiers falsely detected. For certification, we define certified TPR (**CTPR**) as the proportion of a set of attacks that are guaranteed to be detectable, i.e. falling into the certified region.

**Attack Setting:** For each dataset, we create 50 backdoor attacks, each with a randomly selected target class. For each attack, we generate a random trigger, which is a random perturbation embedded

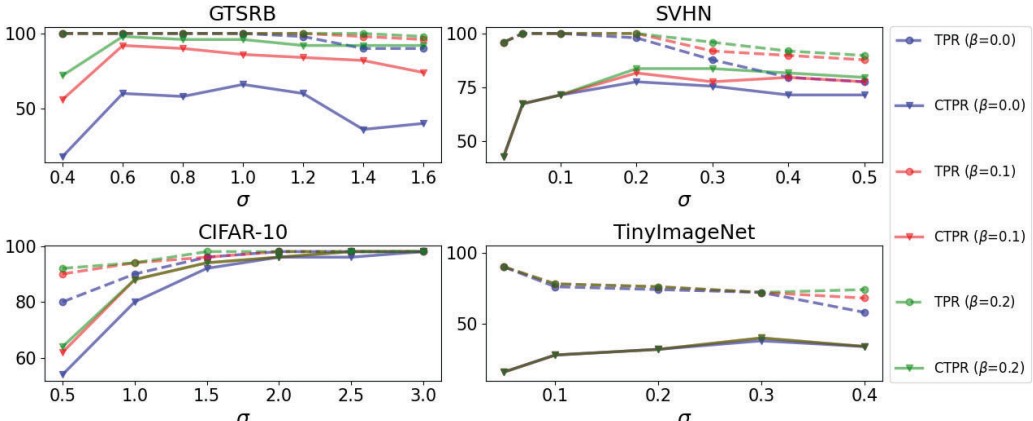

Figure 2: Certification performance of CBD against backdoor attacks with random triggers with perturbation magnitude $\ell_2 \le 0.75$ measured by CTPR (solid) for a range of $\sigma$ for $\beta = 0, 0.1, 0.2$. The CTPRs are all upper-bounded by the TPRs (dashed), showing the correctness of our certification. Notably, CBD achieves up to 98% (100%), 84% (100%), 98% (98%), and 40% (72%) CTPRs (TPRs) on GTSRB, SVHN, CIFAR-10, and TinyImageNet, respectively, across all choices of $\sigma$ and $\beta$. An increment in $\beta$, the assumed ratio of calibration outliers, may lead to further increments in both CTPR and TPR. The hyperparameter $\sigma$ can be determined using the calibration set in practice.

by $\delta(x) = x + v$ with $\|v\|_2 \le 0.75$. The generation process also involves random nullification of the trigger pattern, which helps the trigger to be learned. More details about trigger generation are deferred to App. C.2 due to space limitations. The poisoning ratios for the attacks on GTSRB, SVHN, CIFAR-10, and TinyImageNet are 7.8%, 15.3%, 11.3%, and 12.4%, respectively. Lower poisoning ratios may largely reduce the attack success rate since many randomly generated triggers are relatively hard to learn. Results for triggers with larger perturbation size are shown in App. D.

**Training:** For model architecture, we use the winning model on the leaderboard [31] for GTSRB, MobileNetV2 [54] for SVHN, the same architecture in [82] for CIFAR-10, and ResNet-34 [22] for TinyImageNet. For each dataset, 50 benign models are trained (also on the 40% training set except for GTSRB) to evaluate the FPR. The accuracies for these benign models are roughly 98%, 92%, 78%, and 47% on GTSRB, SVHN, CIFAR-10, and TinyImageNet, respectively. For each attack, we train a model with $\ge 90\%$ attack success rate and $\le 2\%$ degradation in the benign accuracy (or re-generate the attack for training until both conditions are satisfied). More details about the training configurations are shown in App. C.3. Finally, we train 100 shadow models for each of GTSRB, SVHN, and CIFAR-10, and 50 shadow models for TinyImageNet using the same architectures and configurations as above – these shadow models are used by our CBD for detection and certification.

### 5.1.2 Certification Performance

In Fig. 2, we show the CTPR (solid) of our CBD against the attacks with the random perturbation trigger on the four datasets for a range of $\sigma$ and for $\beta = m/N \in \{0, 0.1, 0.2\}$. The TPR (dashed) for each combination of $\sigma$ and $\beta$ is also plotted for reference. Recall that $\beta$ here represents the proportion of assumed outliers in the calibration set for the adjustment of the conformal prediction, and $\sigma$ is the standard deviation of the isotropic Gaussian distribution for the LDP computation. In our experiments, 1024 random Gaussian noises are generated for each sample used to compute the LDP. The significance level for conformal prediction is set to the classical $\alpha = 0.05$ for statistical testing.

In general, our certification is effective, which covers up to 98%, 84%, and 98% backdoor attacks with the random perturbation trigger (all achieved with $\beta = 0.2$) on GTSRB, SVHN, and CIFAR-10, respectively. Even for the very challenging TinyImageNet dataset, CBD certifies up to 40% of these attacks. Moreover, for all choices of $\beta$ and $\sigma$, CTPR is upper-bounded by TPR. For example, for the aforementioned CTPRs on the four datasets, the corresponding TPRs are 100%, 100%, 98%, and 72%, respectively (with FPRs 0%, 0%, 6%, and 10%, respectively, see App. D). In fact, all attacks with the detection guarantee are detected empirically, showing the correctness of our certification.

We also make the following observations regarding the hyperparameters $\beta$ and $\sigma$: 1) An increment in $\beta$ may lead to an increment in both CTPR and TPR. This is due to the existence of a heavy tail

(corresponding to large outliers) in the LDP distribution for the calibration set. While an overly large $\beta$ may cause a significant increment to FPR, our additional results in App. D show that empirically, this is not the case for $\beta \leq 0.2$. 2) Each domain has its own proper range for the choice of $\sigma$. In general, the detection power of CBD (reflected by TPR) reduces as $\sigma$ becomes overly small. This phenomenon agrees with our remarks on Def. 3.1 – SLPV converges to a singleton at the labeled class as $\sigma$ approaches zero, regardless of the presence of a backdoor attack. Moreover, the certification power (reflected by CTPR) also reduces for small $\sigma$'s, which matches the inequality (3) in Thm. 4.1. For overly large $\sigma$'s, on the other hand, LDP for benign classifiers may also grow and possibly approach one, resulting in a large FPR. However, in this case, LDP is no longer computed in a truly 'local' context, contrary to the intuition implied by its name. In Sec. 5.2, we will introduce a practical scheme to choose a proper $\sigma$ for each domain based on the shadow models.

## 5.2 Evaluation of CBD Detection

Here, we show the detection performance of CBD against backdoor attacks with various trigger types, including the BadNet square [18], the "chessboard" (CB) pattern [75], and the blended pattern [8]. For each of GTSRB, SVHN, and CIFAR-10, we train 20 models (using the full training set) for each trigger type following the same training configurations described in Sec. 5.1.1. TinyImageNet is not considered here due to the high training cost. Details for each trigger type and attack settings are shown in App. C.4. We also compare CBD with three state-of-the-art backdoor detectors without certification, which are Neural Cleanse (NC) [67], K-Arm [56], and MNTD [82]. In particular, K-Arm and MNTD require manual selection of the detection threshold. For both of them, we choose the threshold that maximizes the TPR while keeping a 5% FPR for each dataset. Moreover, we devise a "supervised" version of CBD, named $CBD_{sup}$, which still uses LDP as the detection statistic but without the conformal prediction. The detection threshold for $CBD_{sup}$ is determined in the same way as for K-Arm and MNTD, i.e. by maximizing the TPR with a controlled 5% FPR.

In practice, CBD needs to choose a moderately large $\sigma$ for each detection task. To this end, we first initialize a small $\sigma$ such that for each of the $N$ shadow models, the SLPVs for the $K$ samples used for computing the LDP all concentrate at the labeled classes. In this case, the LDPs for all the shadow models are close to $\frac{1}{K}$. Then, we gradually increase $\sigma$ until $\frac{1}{N \times K} \sum_{n=1}^{N} \sum_{k=1}^{K} p_k(x_k^{(n)}|w_n, \sigma) < \psi$ for some relatively small $\psi$, where $x^{(n)}$ is the $k$-th sample for LDP computation for the $n$-th model, i.e. the SLPVs are no longer concentrated at the labeled classes. In the left of Fig. 4, we show the choice of $\sigma$ based on the above scheme for a range of $\psi$, which exhibits a trend of convergence as $\psi$ decreases. We also notice that $\sigma$ selected for $\psi < 0.2$ roughly matches the $\sigma$ choices in Fig. 2 that yields relatively high CTPR and TPR, showing the effectiveness of our scheme. In our evaluation of CBD detection, we set $\psi = 0.1$, which yields $\sigma = 1.15, 0.39, 1.14$ for GTSRB, SVHN, and CIFAR-10, respectively. Other choices of $\sigma$ for $\psi$ less than 0.2 yield similar detection performance.

As shown in Tab. 1, CBD achieves comparable or even higher TPRs than the SOTA detectors (that benefit from unrealistic supervision) for all trigger types on all datasets. CBD also provides non-trivial detection guarantees to most attack types on these datasets. The relatively low CTPRs, e.g. for BadNet on GTSRB, are due to the large perturbation magnitude of the trigger. The even better TPRs achieved by $CBD_{sup}$, though with the same supervision as for the SOTA detectors, show the effectiveness of the LDP statistic in distinguishing backdoored models from benign ones. Such effectiveness is further verified by the receiver operating characteristic (ROC) curves for $CBD_{sup}$. Compared with the baseline detectors, $CBD_{sup}$ achieves generally higher overall areas under curves (AUCs) across the three datasets.

## 5.3 Ablation Study

We show that the time efficiency and data efficiency of CBD can be improved by training fewer shadow models and using fewer samples for training the shadow models, respectively, without significant degradation in the detection or certification performance. In particular, we show in Tab. 2 that CBD with $\beta = 0.2$ achieves similar TPRs and CTPRs for the same set of attacks in Sec. 5.2 when we reduce the number of shadow models from 100 to 50, 25, and 10, respectively. Such robustness of CBD to the calibration size further verifies the clear separation between the benign and backdoored classifiers using our proposed LDP. In Tab. 3, we show that with shadow models trained on only 100

Table 1: Certified detection of CBD for $\beta = 0, 0.1, 0.2$ (shaded), with the empirical detection performance (measured by TPR (%)) compared with NC, K-Arm, MNTD, and CBD$_{sup}$ against BadNet, CB, and Blend attacks on GTSRB, SVHN, and CIFAR-10. The parentheses in each shaded cell contain the CTPR (%) associated with the TPR outside the parentheses. CBD achieves comparable or even higher empirical TPRs compared with the SOTA baselines and provides non-trivial (or even tight) certification for different attacks and datasets. FPRs (%) are reported on benign classifiers.

| | GTSRB | | | | SVHN | | | | CIFAR-10 | | | | Average |
|---|---|---|---|---|---|---|---|---|---|---|---|---|---|
| | benign | BadNet | CB | Blend | benign | BadNet | CB | Blend | benign | BadNet | CB | Blend | TPR |
| NC | 20 | 50 | 75 | 20 | 40 | 80 | 100 | 95 | 20 | 35 | 95 | 60 | 67.8 |
| K-Arm | 5 | 100 | 100 | 100 | 5 | 100 | 70 | 40 | 5 | 100 | 80 | 55 | 82.8 |
| MNTD | 5 | 20 | 0 | 0 | 5 | 10 | 10 | 15 | 5 | 90 | 100 | 75 | 35.6 |
| CBD$_{sup}$ | 5 | 100 | 95 | 100 | 5 | 100 | 100 | 90 | 5 | 65 | 100 | 55 | **89.4** |
| CBD$_0$ | 0 | 75 (5) | 95 (80) | 80 (20) | 0 | 75 (45) | 100 (100) | 80 (75) | 0 | 50 (5) | 100 (90) | 45 (30) | 77.2 |
| CBD$_{0.1}$ | 0 | 90 (15) | 95 (85) | 90 (25) | 0 | 90 (55) | 100 (100) | 80 (80) | 20 | 75 (20) | 100 (95) | 55 (35) | 86.1 |
| CBD$_{0.2}$ | 0 | 90 (15) | 95 (85) | 95 (35) | 0 | 95 (65) | 100 (100) | 90 (80) | 25 | 75 (25) | 100 (100) | 60 (40) | **88.9** |

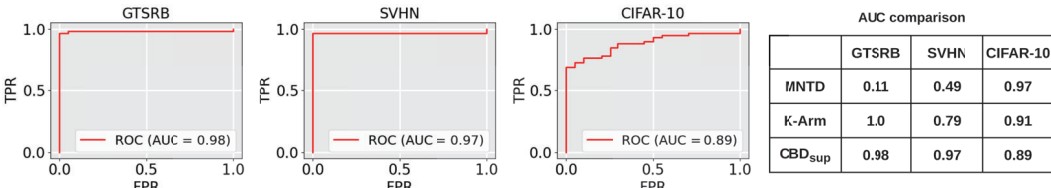

Figure 3: Receiver operating characteristic (ROC) curves of CBD$_{sup}$ aggregated over all three trigger types on GTSRB, SVNH, and CIFAR-10, respectively. CBD$_{sup}$ with our proposed LDP statistic achieves higher overall areas under curves (AUCs) than K-Arm and MNTD across the three datasets.

Table 2: Certified detection of CBD (with CTPR inside the parentheses and TPR outside) for $\beta = 0.2$ using 10, 25, 50, and 100 (the default in Sec. 5.2) shadow models. Both the detection and certification performance of CBD are not significantly affected by reducing the number of shadow models.

| | GTSRB | | | | SVHN | | | | CIFAR-10 | | | |
|---|---|---|---|---|---|---|---|---|---|---|---|---|
| # shadow | benign | BadNet | CB | Blend | benign | BadNet | CB | Blend | benign | BadNet | CB | Blend |
| 10 | 0 | 95 (10) | 95 (85) | 95 (35) | 0 | 75 (45) | 100 (100) | 90 (75) | 25 | 85 (25) | 100 (100) | 55 (40) |
| 25 | 0 | 95 (5) | 95 (85) | 95 (25) | 0 | 95 (45) | 100 (100) | 80 (75) | 25 | 80 (25) | 100 (100) | 70 (40) |
| 50 | 0 | 95 (15) | 95 (85) | 95 (40) | 0 | 95 (65) | 100 (100) | 85 (75) | 25 | 80 (25) | 100 (100) | 60 (40) |
| 100 | 0 | 90 (15) | 95 (85) | 95 (35) | 0 | 95 (65) | 100 (100) | 90 (80) | 25 | 75 (25) | 100 (100) | 60 (40) |

Table 3: Certified detection of CBD (with CTPR inside the parentheses and TPR outside) for $\beta = 0.2$ and $\beta = 0.4$, respectively, with 100 shadow models trained on fewer samples (100 per class) than the default settings in Sec. 5.2. Due to the significantly insufficient training of the shadow models, a larger $\beta$ is needed to maintain a similar level of TPR and CTPR as in Sec. 5.2.

| | GTSRB | | | | SVHN | | | | CIFAR-10 | | | |
|---|---|---|---|---|---|---|---|---|---|---|---|---|
| | benign | BadNet | CB | Blend | benign | BadNet | CB | Blend | benign | BadNet | CB | Blend |
| CBD$_{0.2}$ | 0 | 90 (5) | 95 (85) | 90 (25) | 0 | 75 (45) | 100 (95) | 80 (75) | 0 | 35 (0) | 95 (90) | 45 (40) |
| CBD$_{0.4}$ | 0 | 95 (5) | 95 (85) | 95 (35) | 0 | 95 (45) | 100 (100) | 85 (75) | 5 | 60 (10) | 100 (95) | 55 (40) |

clean samples per class, CBD requires a larger $\beta$ to achieve similar TPRs and CTPRs. This is because more shadow models will exhibit an abnormally large LDP due to significantly insufficient training.

## 5.4 Additional Experiments

**Empirical validation of the stochastic dominance assumption in Thm. 4.2.** In the middle of Fig. 4, we show the histograms (with the associated empirical CDF) of the LDP statistics for the shadow models and the benign models for all four datasets. The statistics for each dataset are obtained using the practically selected $\sigma$. The LDP for the benign models is clearly dominated by the LDP for the shadow models in the sense of first-order stochastic dominance.

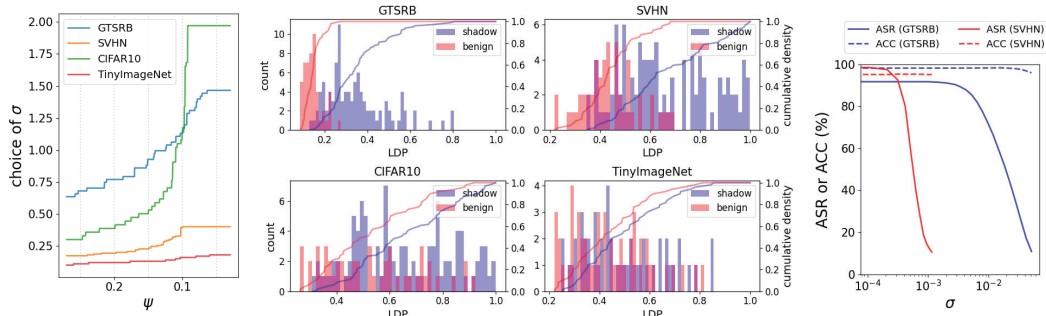

Figure 4: Supportive results: (**Left**) Choice of $\sigma$ for a range of $\psi$ based on our selection scheme, which matches the $\sigma$ choices in Fig. 2 with high CTPR and TPR. (**Middle**) The histograms of the LDP statistics for the shadow models and the benign models, with the associated empirical CDFs. LDP for benign models is stochastically dominated by the LDP for shadow models for all datasets. (**Right**) Vulnerability of WaNet attack on GTSRB and SVHN. Attack success rate (ASR) reduces with a negligible drop in benign accuracy (ACC) when inputs are smoothed by Gaussian noise.

**Class imbalance is not the reason for a large LDP.** In our experiments involving generally balanced datasets, backdoor poisoning (i.e. embedding the trigger in a large variety of samples during training) not only enhances the trigger robustness of the attack, but also introduces an imbalance in the poisoned training set. Thus, it is important to show that the large LDPs observed in our experiments are a result of the robustness of the trigger, rather than the class imbalance. Note that if class imbalance can also cause a large LDP, there will easily be false alarms for benign classifiers trained on imbalanced datasets[1]. Here, we train two groups of benign models on SVHN, with 20 models per group. For the first group, the models are trained on a balanced dataset with 3,000 images per class. For each model in the second group, the training set contains 4,400 additional images labeled to some randomly selected class. With the same model architecture and training configurations, the two groups have similar LDP distributions, with mean±std being 0.445±0.125 and 0.429±0.119, respectively, and with a 0.698 p-value for the t-test for mean. Thus, LDP will not be affected by class imbalance in general, and our method indeed detects the backdoor attack rather than the class imbalance.

**Advanced attacks.** Based on the key ideas in Sec. 1, CBD requires the attack to have a large STR over the sample distribution. However, this requirement is not always satisfied, especially for some advanced attacks with subtle, sample-specific triggers, such as WaNet [51]. Although not detectable or certifiable by our CBD, these attacks can hardly survive noises either from the environment or simple prepossessing-based defenses in practice. To see this, for each of GTSRB and SVHN, we train 20 models with successful WaNet attacks. We consider a simple test-time defense by applying 1024 randomly sampled Gaussian noises from distribution $\mathcal{N}(0, \sigma^2 I)$ to each input and then performing a majority vote. As shown in the right of Fig. 4, for both datasets and for small $\sigma$, the average attack success rate (ASR) quickly drops without clear scarifies in the benign accuracy (ACC).

## 6 Conclusion

In this paper, we proposed CBD, the first certified backdoor detector, which is based on an adjustable conformal prediction using a novel LDP statistic. CBD not only performs detection inference but also provides a condition for attacks that are guaranteed to be detectable. Our theoretical results show that backdoor attacks with a trigger more resilient to noises and with a smaller perturbation magnitude are more likely to be detected with a guarantee. Our empirical results show the strong certification and detection performance of CBD on four benchmark datasets. In future research, we aim to enhance the certification bound to encompass backdoor attacks with larger trigger perturbation norms, such as rotational triggers and subject-based triggers.

**Acknowledgements** This work is partially supported by the National Science Foundation under grant No. 1910100, No. 2046726, No. 2229876, DARPA GARD, the National Aeronautics and Space Administration (NASA) under grant no. 80NSSC20M0229, Alfred P. Sloan Fellowship, and the Amazon research award.

---

[1]This is the case for the backdoor detector in [68] which leverages the overfitting to the backdoor trigger during training – benign classifiers trained on imbalanced datasets will also easily trigger a (false) detection.

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
