## Ethics Statement

The main purpose of this research is to provide the user of DNN classifiers with a method to detect if the model is backdoor attacked without access to the training set. All attacks used to evaluate our detection method in this paper are created by published backdoor attack strategies on public datasets. Thus, we did not create new threats to society. Moreover, our work provides a new perspective on backdoor defense, as it is the first to address the certification of backdoor detection. It helps other researchers to understand the behavior of deep learning systems facing malicious activities. Code is available at `https://github.com/zhenxianglance/CBD`.

## Broader Impacts

While existing backdoor detectors are all empirical [67, 20, 75, 41, 69, 6, 56, 13], our work initiates a new research direction – backdoor detection with certification. Moreover, we first exposed that certified backdoor detectors and certified robustness against backdoor attacks complement each other [86, 71, 27, 53]. Relatively weak attacks, e.g., with a low poisoning rate or a trigger difficult to learn, are more likely to be defeated by robust training strategies with a guarantee. On the other hand, strong backdoor attacks with robustly learned triggers that are resilient to noises are more likely to be detected by our method with a guarantee. We hope that with the development of both branches of certified backdoor defense strategies, future backdoor attacks will be either strong enough to be detected with a guarantee or weak enough to be defeated during training with a guarantee.

## Appendix

## A   Proof of Theorems

### A.1   Proof of Thm. 4.1

The proof of Thm. 4.1 is inspired by the proof of Theorem 1 in [10] which leverages the Neyman-Pearson lemma [49]. In particular, we consider the special case of the lemma for isotropic Gaussian distributions as stated in [10].

**Lemma A.1.** *(Neyman-Pearson for Gaussians with Different Means [10]) Let $X \sim \mathcal{N}(x, \sigma^2 I)$ and $X' \sim \mathcal{N}(x - r, \sigma^2 I)$. Let $h : \mathbb{R}^d \to \{0, 1\}$ be any deterministic or random function. If $\mathcal{A} = \{z \in \mathbb{R} : r^T z \geq \phi\}$ for some $\phi$ and $\mathbb{P}(h(X) = 1) \geq \mathbb{P}(X \in \mathcal{A})$, then $\mathbb{P}(h(X') = 1) \geq \mathbb{P}(X' \in \mathcal{A})$.*

*Proof of Thm. 4.1.* For each $k \in \{1, \cdots, K\}$, we defined $r_k = \delta(x_k) - x_k$ for convenience. Also, assume an isotropic Gaussian noise $\epsilon \sim \mathcal{N}(0, \sigma^2 I)$ with standard deviation $\sigma$. Then, by the definition of STR in Def. 3.2 and the definition of $\pi$ as the minimum STR, we have:

$$R_{\delta,t}(x_k|w, \sigma) = \mathbb{P}(f(x_k + r_k + \epsilon; w) = t) \geq \pi$$

Now, we define a half space $\mathcal{A}_k = \{x : r_k^T(x - x_k - r_k) \geq \sigma||r_k||_2 \Phi^{-1}(1 - \pi)\}$, such that:

$$
\begin{aligned}
\mathbb{P}(x_k + r_k + \epsilon \in \mathcal{A}_k) &= \mathbb{P}(r_k^T \epsilon \geq \sigma||r_k||_2 \Phi^{-1}(1 - \pi)) \\
&= \mathbb{P}(\sigma||r_k||_2 Z \geq \sigma||r_k||_2 \Phi^{-1}(1 - \pi)) && \triangleright Z \sim \mathcal{N}(0, 1) \\
&= 1 - \Phi(\Phi^{-1}(1 - \pi)) \\
&= \pi
\end{aligned}
$$

Thus:

$$\mathbb{P}(f(x_k + r_k + \epsilon; w) = t) \geq \mathbb{P}(x_k + r_k + \epsilon \in \mathcal{A}_k)$$

and based on Lemma A.1, we have:

$$\mathbb{P}(f(x_k + \epsilon; w) = t) \geq \mathbb{P}(x_k + \epsilon \in \mathcal{A}_k) \tag{4}$$

Based on the definition of LDP in Def. 3.3, and for the given samples $x_1, \cdots, x_k$ used for LDP computation, we have:

$$s(w) = ||\frac{1}{K}\sum_{k=1}^{K} \boldsymbol{p}(x_k|w, \sigma)||_\infty$$

$$\geq \frac{1}{K}\sum_{k=1}^{K} p_t(x_k|w, \sigma)$$

$$= \frac{1}{K}\sum_{k=1}^{K} \mathbb{P}(f(x_k + \epsilon; w) = t) \qquad\qquad \triangleright \text{Def. 3.1}$$

$$\geq \frac{1}{K}\sum_{k=1}^{K} \mathbb{P}(r_k^T(\epsilon - r_k) \geq \sigma||r_k||_2 \Phi^{-1}(1 - \pi)) \qquad\qquad \triangleright \text{Inequality (4)}$$

$$= \frac{1}{K}\sum_{k=1}^{K} \mathbb{P}(Z \geq \Phi^{-1}(1 - \pi) + \frac{||r_k||_2}{\sigma})$$

$$\geq \frac{1}{K}\sum_{k=1}^{K} \mathbb{P}(Z \geq \Phi^{-1}(1 - \pi) + \frac{\Delta}{\sigma}) \qquad\qquad \triangleright \Delta = \max_{k=1,\cdots,K} ||r_k||_2$$

$$= 1 - \Phi(\Phi^{-1}(1 - \pi) + \frac{\Delta}{\sigma})$$

Since the attack is detected if $q_m(w) \leq \alpha$, which is true when $s(w) > s_{(N-m-\lfloor\alpha(N-m+1)\rfloor)}$ is satisfied based on Eq. (2). Thus, the detection of the attack is guaranteed if inequality (3) holds. $\square$

## A.2  Proof of Thm. 4.2

*Proof of Thm. 4.2.* Let $s_{(1)}, \cdots, s_{(N)}$ be the order statistics associated with $s_1, \cdots, s_N$ and denote $S = s(W)$ for simplicity. Then, the false positive rate can be represented as the following:

$$\mathbb{P}(q_m(W) \leq \alpha|\mathcal{S}_N)$$
$$=\mathbb{P}(\min\{|\{s \in \mathcal{S}_N : s < S\}|, N - m\} \geq N - m - \alpha(N - m + 1)|\mathcal{S}_N) \qquad \triangleright \text{Eq. (2)}$$
$$=\mathbb{P}(s_{(N-m)} < S) + \mathbb{P}(s_{(N-m-l)} < S \leq s_{(N-m)})$$
$$=1 - \tilde{F}(s_{(N-m-l)})$$

For any $z \in [0, 1]$, we have:

$$\mathbb{P}(1 - \tilde{F}(s_{(N-m-l)}) \leq z)$$
$$=\mathbb{P}(s_{(N-m-l)} \geq \tilde{F}^{-1}(1 - z))$$
$$\geq\mathbb{P}(s_{(N-m-l)} \geq F^{-1}(1 - z)) \qquad\qquad \triangleright F \text{ dominates } \tilde{F} \Rightarrow \tilde{F}^{-1} \text{ dominates } F^{-1}$$
$$=1 - \sum_{j=N-m-l}^{N} \binom{N}{j}(1 - z)^j z^{N-j} \qquad\qquad \triangleright \text{Distribution of order statistic}$$
$$= \sum_{j=0}^{N-m-l-1} \binom{N}{j}(1 - z)^j z^{N-j} \qquad\qquad \triangleright \text{Binomial expansion}$$
$$=I_{(z)}(m + l + 1, N - m - l) \qquad\qquad \triangleright \text{CDF of Binomial distribution}$$

where $I_{(z)}(a, b)$ represents the incomplete Beta function, which is also the CDF of the Beta distribution $\text{Beta}(a, b)$. Thus, the distribution of the false positive rate of our CBD is dominated by a Beta distribution $\text{Beta}(m + l + 1, N - m - l)$ with $l = \lfloor\alpha(N - m + 1)\rfloor$ in the sense of first-order stochastic dominance. $\square$

## A.3 Proof of Col. 4.3

*Proof of Col. 4.3.* Based on Thm. 4.2, we have

$$\mathbb{P}(Z_N \leq \alpha + (1-\alpha)\beta + \xi) \geq \mathbb{P}(B \leq \alpha + (1-\alpha)\beta + \xi) \tag{5}$$

where $B$ follows a Beta distribution $\text{Beta}(m+l+1, N-m-l)$ with $l = \lfloor \alpha(N-m+1) \rfloor$. Thus, the mean and variance of of $B$ are:

$$\mathbb{E}[B] = \frac{m+l+1}{N+1}$$

$$\text{Var}[B] = \frac{(m+l+1)(N-m-l)}{(N+1)^2(N+2)}$$

By Chebyshev's inequality:

$$\mathbb{P}(|B - \frac{m+l+1}{N+1}| \geq \xi) \leq \frac{1}{\xi^2}\frac{(m+l+1)(N-m-l)}{(N+1)^2(N+2)} \tag{6}$$

Note that $l = \lfloor \alpha(N-m+1) \rfloor \in [\alpha(N-m+1), \alpha(N-m+1)+1)$. Thus, algebra shows that:

$$\mathbb{P}(B \geq \alpha + (1-\alpha)\beta + \xi + \frac{\alpha+1}{N})$$

$$=\mathbb{P}(B \geq \frac{m+\alpha(N-m+1)+1}{N} + \xi)$$

$$\leq\mathbb{P}(B \geq \frac{m+l+1}{N+1} + \xi)$$

$$\leq\mathbb{P}(|B - \frac{m+l+1}{N+1}| \geq \xi)$$

and

$$\frac{1}{\xi^2(N+2)}(\beta + \alpha(1-\beta+\frac{1}{N}) + \frac{1}{N})(1-\beta-\alpha(1-\beta+\frac{1}{N})+\frac{1}{N})$$

$$=\frac{1}{\xi^2}\frac{(m+\alpha(N-m+1)+1)(N-m-\alpha(N-m+1)+1)}{N^2(N+2)}$$

$$\geq\frac{1}{\xi^2}\frac{(m+l+1)(N-m-l)}{(N+1)^2(N+2)}$$

According to inequality (6) and also because of the continuity of the Beta distribution:

$$\lim_{N \to +\infty} \mathbb{P}(B \leq \alpha + (1-\alpha)\beta + \xi)$$

$$= \lim_{N \to +\infty} \mathbb{P}(B \leq \alpha + (1-\alpha)\beta + \xi + \frac{\alpha+1}{N})$$

$$=1 - \lim_{N \to +\infty} \mathbb{P}(B \geq \alpha + (1-\alpha)\beta + \xi + \frac{\alpha+1}{N})$$

$$\geq1 - \lim_{N \to +\infty} \frac{1}{\xi^2(N+2)}(\beta + \alpha(1-\beta+\frac{1}{N}) + \frac{1}{N})(1-\beta-\alpha(1-\beta+\frac{1}{N})+\frac{1}{N})$$

$$=1$$

Thus, the corollary is proved by applying the sandwich theorem to inequality (5). $\qquad\square$

## B Stochastic Dominance Assumption in Thm. 4.2

In Thm. 4.2, we assume that the distribution $F$ for the LDP of the benign shadow models with inadequate training dominates the distribution $\tilde{F}$ for the LDP of the actual benign classifiers trained on more abundant data in the sense of first-order stochastic dominance. This assumption holds in general in practice. It is widely believed that models are trained to learn the data distribution [26]. Thus, intuitively, training with insufficient sampling of the data distribution will not only lead to

a high empirical loss but also result in a poor robustness on the test instances. In our setting, this is reflected by a small samplewise local probability for the labeled class for most samples used for computing LDP, which may easily lead to a large LDP. In the following, we show that a larger deviation of the learned decision boundary of a binary Bayesian classifier will affect its LDP. The analysis can be extended to multi-class scenarios based on the "winner takes all" rule. Some notations may be abused but are all constrained to the current section.

**Theorem B.1.** *Consider two classes $\{a, b\}$ with equal prior and arbitrary sample distributions $\mu_a$ and $\mu_b$, respectively. Consider a Bayes classifier $g(z) = \arg\max_{k \in \{a,b\}} \mu_k(z)$ (i.e. with a decision boundary $\frac{\mu_a(z)}{\mu_b(z)} = 1$). We arbitrarily assume that the expected LDP for $g$ is associated with class $b$:*

$$s = \frac{1}{2}(\mathbb{E}_{X \sim \mu_a}[\mathbb{P}(g(X + \epsilon) = b)] + \mathbb{E}_{X \sim \mu_b}[\mathbb{P}(g(X + \epsilon) = b)]) \tag{7}$$

*Moreover, consider any two classifiers $g_1$ and $g_2$ with decision boundaries $\frac{\mu_a(z)}{\mu_b(z)} = t_1$ and $\frac{\mu_a(z)}{\mu_b(z)} = t_2$, respectively. Then, if $t_2 \geq t_1 \geq 1$, the expected LDP for the two classifiers, denoted by $s_1$ and $s_2$, respectively, will satisfy $s_2 \geq s_1 \geq s$, i.e. the more deviation in the decision boundary from the optimum, the larger expected LDP.*

*Proof.* For each $i \in \{1, 2\}$, the expectation of the LDP for classifier $g_i$ can be expressed as:

$$s_i = \max_{k \in \{a,b\}} \frac{1}{2}(\mathbb{E}_{X \sim \mu_a}[g_i(X + \epsilon) = k] + \mathbb{E}_{X \sim \mu_b}[g_i(X + \epsilon) = k]) \tag{8}$$

Since $t_2 \geq t_1 \geq 0$, given the decision boundaries of $g_1$ and $g_2$, we have:

$$\mathbb{E}_{X \sim \mu_a}[\mathbb{P}(g_1(X + \epsilon) = a)]$$

$$= \int \mathbb{P}(\frac{\mu_a(z + \epsilon)}{\mu_b(z + \epsilon)} > t_1)\mu_a(z)dz$$

$$= \int \left[\mathbb{P}(\frac{\mu_a(z + \epsilon)}{\mu_b(z + \epsilon)} > t_2) + \mathbb{P}(t_1 < \frac{\mu_a(z + \epsilon)}{\mu_b(z + \epsilon)} \leq t_2)\right]\mu_a(z)dz$$

$$= \mathbb{E}_{X \sim \mu_a}[\mathbb{P}(g_2(X + \epsilon) = a)] + \int \mathbb{P}(t_1 < \frac{\mu_a(z + \epsilon)}{\mu_b(z + \epsilon)} \leq t_2)\mu_a(z)dz$$

and similarly,

$$\mathbb{E}_{X \sim \mu_b}[\mathbb{P}(g_2(X + \epsilon) = b)]$$

$$= \int \mathbb{P}(\frac{\mu_a(z + \epsilon)}{\mu_b(z + \epsilon)} \leq t_2)\mu_b(z)dz$$

$$= \int \left[\mathbb{P}(\frac{\mu_a(z + \epsilon)}{\mu_b(z + \epsilon)} \leq t_1) + \mathbb{P}(t_1 < \frac{\mu_a(z + \epsilon)}{\mu_b(z + \epsilon)} \leq t_2)\right]\mu_b(z)dz$$

$$= \mathbb{E}_{X \sim \mu_b}[\mathbb{P}(g_1(X + \epsilon) = b)] + \int \mathbb{P}(t_1 < \frac{\mu_a(z + \epsilon)}{\mu_b(z + \epsilon)} \leq t_2)\mu_b(z)dz$$

According to above relations between $g_1$ and $g_2$, we have

$$\frac{1}{2}(\mathbb{E}_{X \sim \mu_a}[\mathbb{P}(g_2(X + \epsilon) = b)] + \mathbb{E}_{X \sim \mu_b}[\mathbb{P}(g_2(X + \epsilon) = b)])$$

$$= \frac{1}{2}(1 - \mathbb{E}_{X \sim \mu_a}[\mathbb{P}(g_2(X + \epsilon) = a)] + \mathbb{E}_{X \sim \mu_b}[\mathbb{P}(g_2(X + \epsilon) = b)])$$

$$= \frac{1}{2}\left(1 - \mathbb{E}_{X \sim \mu_a}[\mathbb{P}(g_1(X + \epsilon) = a)] + \int \mathbb{P}(t_1 < \frac{\mu_a(z + \epsilon)}{\mu_b(z + \epsilon)} \leq t_2)\mu_a(z)dz\right.$$

$$\left. + \mathbb{E}_{X \sim \mu_b}[\mathbb{P}(g_1(X + \epsilon) = b)] + \int \mathbb{P}(t_1 < \frac{\mu_a(z + \epsilon)}{\mu_b(z + \epsilon)} \leq t_2)\mu_b(z)dz)\right)$$

$$= \frac{1}{2}(\mathbb{E}_{X \sim \mu_a}[\mathbb{P}(g_1(X + \epsilon) = b)] + \mathbb{E}_{X \sim \mu_b}[\mathbb{P}(g_1(X + \epsilon) = b)])$$

$$+ \frac{1}{2}\int \mathbb{P}(t_1 < \frac{\mu_a(z + \epsilon)}{\mu_b(z + \epsilon)} \leq t_2)(\mu_a(z) + \mu_b(z))dz$$

$$\geq \frac{1}{2}(\mathbb{E}_{X \sim \mu_a}[\mathbb{P}(g_1(X + \epsilon) = b)] + \mathbb{E}_{X \sim \mu_b}[\mathbb{P}(g_1(X + \epsilon) = b)])$$

Similarly, since $t_1 \geq 1$, for $g_1$ and $g$, we have:

$$\frac{1}{2}\left(\mathbb{E}_{X \sim \mu_a}[\mathbb{P}(g_1(X + \epsilon) = b)] + \mathbb{E}_{X \sim \mu_b}[\mathbb{P}(g_1(X + \epsilon) = b)]\right)$$

$$\geq \frac{1}{2}\left(\mathbb{E}_{X \sim \mu_a}[\mathbb{P}(g(X + \epsilon) = b)] + \mathbb{E}_{X \sim \mu_b}[\mathbb{P}(g(X + \epsilon) = b)]\right)$$

Thus, $s_2 \geq s_1 \geq s$ if $t_2 \geq t_1 \geq 1$. □

## C   Details for the Experimental Setup

### C.1   Details for the Datasets

GTSRB is an image dataset for German traffic signs from 43 classes [60]. The training set and the test set contain 39,209 and 12,630 images, respectively. The image sizes vary in a relatively large range. Thus, we resize all the images to $32 \times 32$ in our experiments for convenience.

SVHN is a real-world image dataset created from house numbers in Google Street View images. It contains $32 \times 32$ color images from the 10 classes of the ten digits [47]. The training set and the test set contain 73,257 and 26,032 images, respectively.

CIFAR-10 is a benchmark dataset with $32 \times 32$ color images from 10 classes for different categories of objects [30]. The training set contains 50,000 images, and the test set contains 10,000 images, both evenly distributed in the 10 classes.

TinyImageNet is a subset of the ImageNet dataset [11]. It contains color images from 200 classes downsized to $64 \times 64$. The training set and the test set contain 100,000 and 10,000 images, respectively, both evenly distributed into the 200 classes.

### C.2   Details for Trigger Generation in Sec. 5.1.1

The triggers used by the attacks to evaluate the certification of CBD are generated with randomness. But for each attack, once the trigger is generated, the same trigger will be used by all the poisoned images. For the trigger pattern on GTSRB, each pixel is randomly perturbed with a 0.25 probability. For each pixel being perturbed, the perturbation sizes for all three channels are randomly and independently selected in the interval [0, 12/255]. For the trigger pattern on SVHN, each pixel among the $16 \times 16$ patch on the top left of the image is randomly perturbed with a 0.5 probability. The perturb sizes for all three channels are randomly and independently selected between -9/255 and 9/255. For the trigger pattern on CIFAR-10, we found that purely random patterns are too hard to learn, even with the help of nullification. Thus, we use the recursive pattern that looks like a chessboard in [75], but with the perturbation size randomly selected from 1/255, 2/255, 3/255, and 4/255. For TinyImageNet, we used the same trigger generation approach as for SVHN. For all the patterns, we apply a projection to ensure that the $\ell_2$ norm of the trigger perturbation is no larger than 0.75.

### C.3   Detailed Training Configurations

For GTSRB, training is performed for 100 epochs with a batch size of 128 and a learning rate of $10^{-3}$ (with 0.1 decay per 20 epochs), using the Adam optimizer [28]. Each training image is also augmented with a $\pm 5$ degree random rotation.

For SVHN, training is performed for 50 epochs with a batch size of 128 and a learning rate of $10^{-3}$, also using the Adam optimizer. Each training image is augmented with a random horizontal flipping.

For CIFAR-10, training is performed for 100 epochs with a batch size of 32 and a learning rate of $10^{-3}$, using the Adam optimizer. Training data augmentation involves random horizontal flipping and random cropping.

For TinyImageNet, training is performed for 100 epochs with a batch size of 128 and a learning rate of $10^{-3}$, using the Adam optimizer. Training data augmentation involves random horizontal flipping and random cropping.

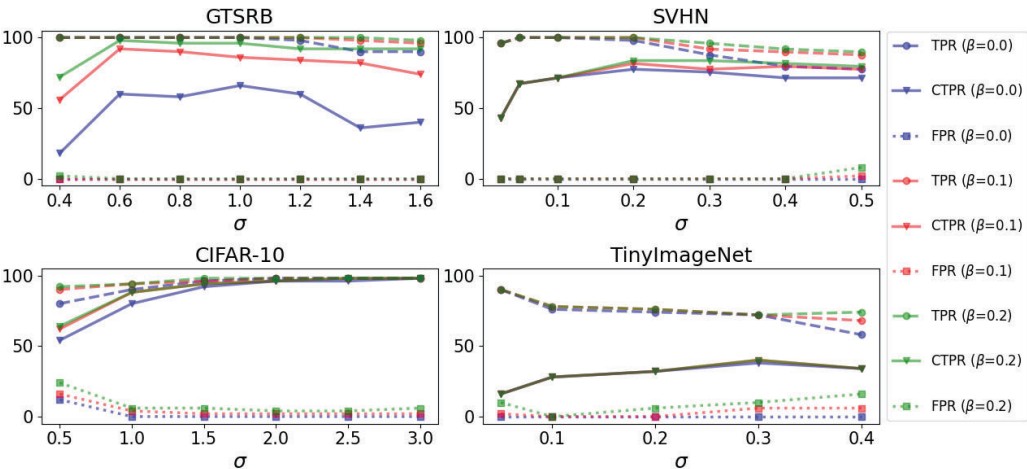

Figure 5: CTPR (solid) of CBD against backdoor attacks with random perturbation triggers with perturbation magnitude $\ell_2 \leq 0.75$ over a range of $\sigma$ for $\beta = m/N \in \{0, 0.1, 0.2\}$ for the four datasets. TPR (dashed) and FPR (dotted) are also plotted for reference.

### C.4  Details for the Triggers Uses in Sec. 5.2

For the BadNet trigger in [18], we use a $2 \times 2$ noisy patch with a randomly selected location for each attack on GTSRB and CIFAR-10. For SVHN, we increase the patch size to $3 \times 3$ to ensure the trigger is learned. The poisoning ratios on GTSRB, SVHN, and CIFAR-10 are 7.8%, 0.4%, and 3%, respectively.

For the "chessboard" pattern in [75], we use 3/255 maximum perturbation magnitude for SVHN and CIFAR-10. For GTSRB, we use a larger perturbation magnitude 6/255 to ensure the trigger is learned. The poisoning ratios on GTSRB, SVHN, and CIFAR-10 are 7.8%, 2.7%, and 6%, respectively.

For the blend trigger in [8], we generate an image-wide noisy pattern as the trigger, with a mixing rate of 0.1. The poisoning ratios on GTSRB, SVHN, and CIFAR-10 are 7.8%, 2.7%, and 3%, respectively.

## D  Additional Certification Results

In Sec. 5.1, we focused on backdoor attacks with random perturbation triggers with perturbation magnitude $\ell_2 \leq 0.75$. In Fig. 2, we showed the CTPR and TPR for CBD to validate our detection guarantee on backdoored models. In particular, we showed that CBD achieves maximum CTPRs (TPRs) 98% (100%), 84% (100%), 98% (98%), and 40% (72%) on GTSRB, SVHN, CIFAR-10, and TinyImageNet, respectively, across all choices of $\sigma$ and $\beta$. Here, in Fig. 5, we also show the FPR for all combinations of $\sigma$ and $\beta$. Especially, the FPRs corresponding to the aforementioned TPRs and CTPRs on the four datasets are 0%, 0%, 6%, and 10%, respectively. From the figure, we also observe that for $\beta \leq 0.2$, CTPR and/or TPR may be improved by the increment in $\beta$, with a rather small sacrifice in FPR.

Also, in this section, we consider triggers with larger perturbation magnitudes with $0.75 < \ell_2 \leq 1.5$. The experiments follow the same settings as described in Sec. 5.1.1. The $\ell_2$ constraint is satisfied by increasing the per-pixel perturbation sizes accordingly. For GTSRB, for example, the trigger is generated in the same way as described in App. C.2, but for each pixel being perturbed, the perturbation sizes for all three channels are randomly and independently selected in the interval [12/255, 24/255]. In Fig. 6, we show the CTPR for the same range of $\sigma$ for each dataset as in Fig. 5, and for $\beta = 0, 0.1, 0.2$. The TPR and the FPR are also plotted for reference. Compared with the performance of CBD for triggers with smaller perturbation magnitudes, there is a drop in both CTPR and TPR, especially for GTSRB and SVHN. However, CBD still achieves up to 100% (68%), 100% (76%), 100% (100%), and 84% (28%) empirical (certified) true positive rates on GTSRB, SVHN, CIFAR-10, and TinyImageNet, respectively.

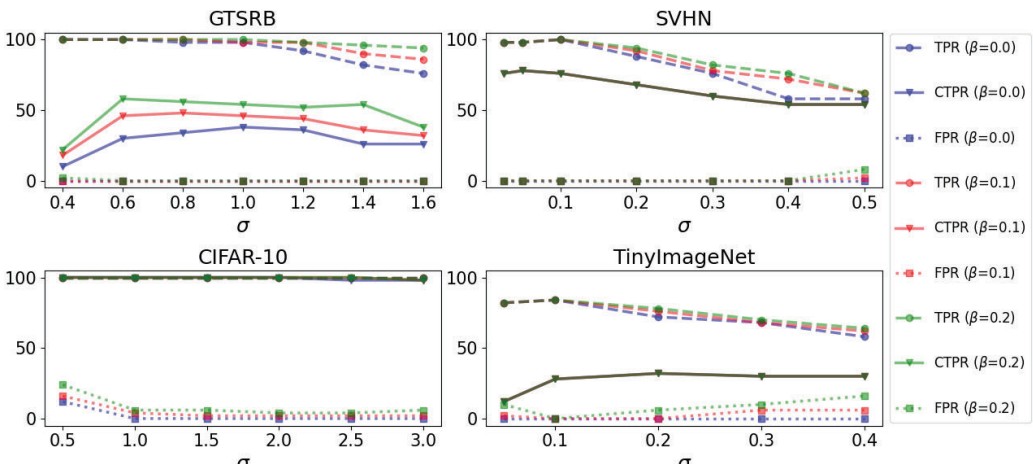

Figure 6: CTPR (solid) of CBD against backdoor attacks with random triggers with perturbation magnitude $0.75 < \ell_2 \leq 1.5$ over a range of $\sigma$ for $\beta = m/N \in \{0, 0.1, 0.2\}$ for the four datasets. TPR (dashed) and FPR (dotted) are also plotted for reference.

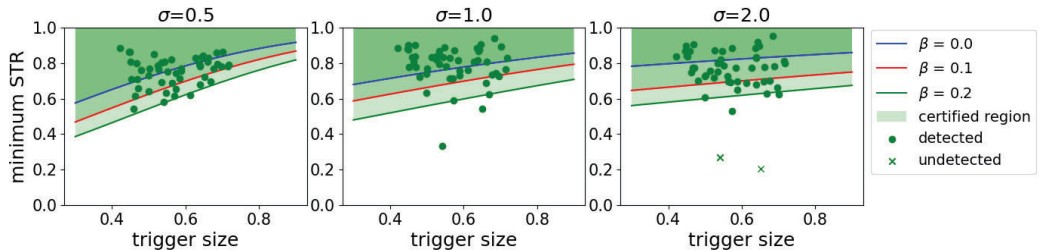

Figure 7: Example of the certified region obtained using Thm. 4.1 on GTSRB. Most of the attacks with random triggers with magnitude constraint $\ell_2 \leq 0.75$ fall into the certified region for $\beta = 0.2$.

In Fig. 7 and Fig. 8, we show the example certified regions on GTSRB for $\sigma = 0.5, 1.0, 2.0$ and for $\beta = 0, 1.0, 2.0$. The certified regions in the two sets of figures are obtained from Thm. 4.1 though plotted for different ranges of the trigger size. For all combinations of $\sigma$ and $\beta$, the shape of the certified region matches our theoretical result that attacks with a larger STR and a smaller perturbation size of the trigger are more likely to be detected with a guarantee. Especially for a larger trigger size, the detection guarantee also requires a larger STR, resulting in a reduced margin for the certified region as the trigger size grows.

In Fig. 7, we show that empirically, most of the attacks with the random perturbation trigger satisfying the $\ell_2 \leq 0.75$ constraint fall into the certified region associated with $\beta = 0.2$, showing the strong detection and certification power of our CBD against attacks with moderate-sized triggers. In Fig. 8, for larger trigger perturbation magnitudes, there is a drop in the number of attacks falling into the certified region, leading to a drop in the CTPR. Although these results show the limitation of CBD certification against triggers with overly large perturbation magnitudes, these triggers may be easily revealed by other (even simpler) detectors, including a human inspection. Moreover, we found that sometimes, the minimum STR of an attack increases as the perturbation magnitude of the trigger increases. The same phenomenon is also observed on CIFAR-10, which allows attacks with even larger trigger perturbation magnitudes to be detected and certified by our CBD. One possible reason is that for triggers with a small perturbation magnitude, the class discriminate features in the samples embedded with the trigger may compromise the learning of the trigger during training.

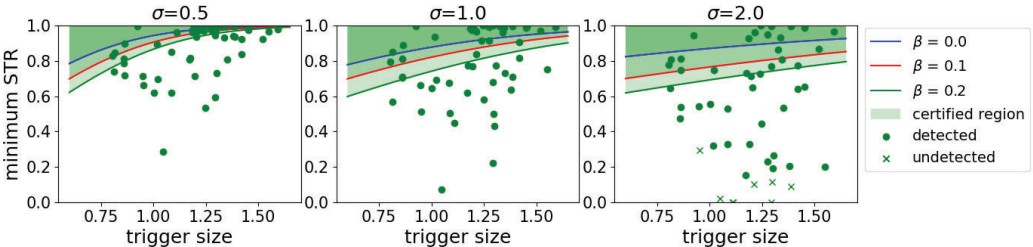

Figure 8: Example of the certified region obtained using Thm. 4.1 on GTSRB (plotted for a different range of trigger sizes compared with Fig. 7). There are still many attacks with random triggers with magnitude constraint $0.75 < \ell_2 \le 1.5$ fall into the certified region.

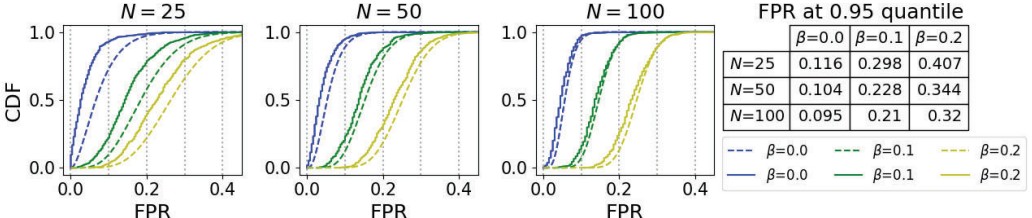

Figure 9: Validation of the FPR upper bound in Thm. 4.2 using the shadow models on GTSRB. For each calibration size $N$ and $\beta$, the empirical CDF of the FPR (solid) is indeed dominated by the CDF of the Beta random variable (dashed) derived by Thm. 4.2. In the table on the right, the decrement of FPR at 0.95 quantiles as $N$ grows implies a better FPR control with a larger calibration set.

## E   Empirical Validation of the FPR Guarantee

We validate the probabilistic upper bound for the FPR in Thm. 4.2 using the shadow models on GTSRB as an example. We train 300 shadow models and sample a number of them to form the calibration set to evaluate the FPR on the remaining ones. Such a process is repeated 500 times for each calibration size $N \in \{25, 50, 100\}$ and $\beta = m/N \in \{0, 0.1, 0.2\}$, which produces an empirical CDF for each combination of $N$ and $\beta$, as shown in Fig. 9. Since in the settings here, the LDP distribution for the classifiers to be inspected is the same as the distribution of the LDPs in the calibration set, the stochastic dominance assumption in Thm. 4.2 is satisfied. Thus, for each $N$ and $\beta$, the empirical CDF (solid) is dominated by the CDF of the Beta random variable obtained by Thm. 4.2 (dashed) as shown in Fig. 9. Moreover, we find that for all $\beta$ choices, the FPR value at 0.95 quantile decreases as $N$ increases, which implies that a larger calibration set (which requires more shadow models) may help to reduce the worst cases FPR.

## F   Computational Complexity

For a new domain, our detection procedure requires $N$ shadow model training and $(N + 1)$ LDP estimation. The $N$ shadow models and their estimated LDP statistics can also be used in the future for the same domain. On a single RTX 2080 Ti card, training of one shadow model on GTSRB, SVHN, CIFAR-10, and TinyImageNet requires 39s, 45s, 31s, and 2462s, respectively, while LDP estimation requires 8s, 4s, 7s, and 86s on the four datasets respectively.

## G   Additional Technical Details for CBD Detection and Certification

In Sec. 3.4, we described the detection procedure of CBD, which employs an adjustable conformal prediction that assumes $m$ overly large outliers in the calibration set, where $m$ may be prescribed based on prior knowledge. Here, we introduce a simple scheme based on anomaly detection that can be used to determine $m$ in practice with very little prior knowledge. Let $\mathcal{S}_N = \{s_1, \cdots, s_N\}$ be the calibration set of LDP statistics obtained from the shadow models. Our goal is to check if there are

large outliers. To this end, we compute the median absolute deviation (MAD [21]) over the elements in $\mathcal{S}_N$ by $\mathrm{MAD} = \mathrm{median}_{s \in \mathcal{S}_N}(|s - \mathrm{median}_{s' \in \mathcal{S}_N}(s')|)$. Under the Gaussian assumption, an outlier $s$ for 0.05 single-tailed significance level will satisfy $(s - \mathrm{median}_{s' \in \mathcal{S}_N}(s'))/(1.4826 \cdot \mathrm{MAD}) \geq 1.645$. Using this scheme, we obtain the proportion of outliers $\beta = m/N$ as 0.16, 0.03, 0.01, and 0.06 for GTSRB, SVHN, CIFAR-10, and TinyImageNet, respectively. While the scheme suggests relatively conservative adjustment to the conformal prediction for most datasets, the suggested choice of $\beta$ for GTSRB will lead to a clear increment in CTPR with negligible increment in FPR, based on our results in Fig. 5. Other anomaly detection methods with more prior knowledge may also be used.

In this section, we also present the detailed algorithm for CBD detection and certification.

---

**Algorithm 1** Algorithm for CBD detection

---

1: **Input:** classifier $f(\cdot; w)$ to be inspected with $K$ classes; clean validation set $\mathcal{D}$ for detection.
2: **Hyperparameter:** standard devistion $\sigma$ for the Gaussian noise; size of the calibration set $N$; number of assumed outliers in the calibration set $m$; significance level $\alpha$ for conformal prediction; $J$ random samples of the Gaussian noise ($J = 1024$ in our experiments).
3: **Step 1: estimate LDP for $f(\cdot; w)$.**
4: Randomly sample $x_1, \cdots x_K$ from $\mathcal{D}$ satisfying $f(x_k; w) = k$ for $\forall k \in \{1, \cdots, K\}$.
5: **for** $k = 1 : K$ **do**
6:      Initialize a frequency vector $\boldsymbol{q} = \boldsymbol{0}$
7:      **for** $j = 1 : J$ **do**
8:          Sample $\epsilon$ from $\mathcal{N}(0, \sigma^2 I)$
9:          Get the class decision $y = f(x_k + \epsilon; w)$
10:          $[\boldsymbol{q}]_y \leftarrow [\boldsymbol{q}]_y + 1$
11:      Compute SLPV for $x_k$ by: $\boldsymbol{p}(x_k|w, \sigma) = \frac{\boldsymbol{q}}{||\boldsymbol{q}||_1}$
12: Compute LDP for $f(\cdot; w)$ by: $s(w) = ||\frac{1}{K} \sum_{k=1}^{K} \boldsymbol{p}(x_k|w, \sigma)||_\infty$
13: **Step 2: construct a calibration set $\mathcal{S}_N$.**
14: Train shadow models $f(\cdot; w_1), \cdots, f(\cdot; w_N)$ on randomly sampled $\mathcal{D}_{\mathrm{Train}} \subset \mathcal{D}$
15: **for** $n = 1 : N$ **do**
     Randomly sample $x_1^{(n)}, \cdots, x_K^{(n)}$ from $\mathcal{D} \setminus \mathcal{D}_{\mathrm{Train}}$ satisfying $f(x_k^{(n)}; w_n) = k$ for $\forall k \in \{1, \cdots, K\}$.
16:      **for** $k = 1 : K$ **do**
17:          Initialize a frequency vector $\boldsymbol{q} = \boldsymbol{0}$
18:          **for** $j = 1 : J$ **do**
19:              Sample $\epsilon$ from $\mathcal{N}(0, \sigma^2 I)$
20:              Get the class decision $y = f(x_k^{(n)} + \epsilon; w_n)$
21:              $[\boldsymbol{q}]_y \leftarrow [\boldsymbol{q}]_y + 1$
22:          Compute SLPV for $x_k^{(n)}$ by: $\boldsymbol{p}(x_k^{(n)}|w_n, \sigma) = \frac{\boldsymbol{q}}{||\boldsymbol{q}||_1}$
23:      Compute LDP for $f(\cdot; w_n)$ by: $s(w_n) = ||\frac{1}{K} \sum_{k=1}^{K} \boldsymbol{p}(x_k^{(n)}|w_n, \sigma)||_\infty$
24: Construct calibration set $\mathcal{S}_N = \{s(w_1)), \cdots, s(w_N)\}$.
25: **Step 3&4: adjustable conformal prediction.**
26: Compute p-value $q_m(w) = 1 - \frac{1 + \min\{|\{s \in \mathcal{S}_N : s < s(w)\}|, N - m\}}{N - m + 1}$.
27: **Output:** $f(\cdot; w)$ is backdoored if $q_m(w) \leq \alpha$; otherwise, $f(\cdot; w)$ is not backdoored.

---

# H    Effectiveness of Backdoor Attacks with Random Perturbation Trigger

In Sec. 5.1, we evaluated the certification performance of our CBD against backdoor attacks with the random perturbation trigger. The attacks were implemented following the conventional strategy by poisoning the training set with some images embedded with the trigger and labeled to the backdoor target class. We stated that attacks with poisoning rate lower than those used by our experiments will easily lead to a failed attack.

Here, we show more details regarding the effectiveness of these attacks. In particular, we focus on the CIFAR-10 dataset as an example. In Sec. 5.1, we used a 11.3% poisoning rate (250 poisoned images from 9 classes other than the backdoor target class, divided by 20000 training images). Here, we also consider attacks with 50, 100, 150, and 200 poisoned images per class, i.e. with 2.3%, 4.5%,

---

**Algorithm 2** Algorithm for CBD Certification

---

1: **Input:** backdoor trigger $\delta$; target class $t$; victim classifier $f(\cdot; w)$ with $K$ classes; samples $x_1, \cdots, x_K$ for detection; calibration set $\mathcal{S}_N$; a standard Gaussian CDF $\Phi$.
2: **Hyperparameter:** standard deviation $\sigma$ for the Gaussian noise; size of the calibration set $N$; number of assumed outliers in the calibration set $m$; significance level $\alpha$ for conformal prediction; $J$ random samples of the Gaussian noise ($J = 1024$ in our experiments).
3: **for** $k = 1 : K$ **do**
4:     Initialize $r = 0$
5:     **for** $j = 1 : J$ **do**
6:         Sample $\epsilon$ from $\mathcal{N}(0, \sigma^2 I)$
7:         **if** $f(\delta(x_k) + \epsilon; w) = t$ **then**
8:             $r \leftarrow r + 1$
9:         STR for $x_k$ is $R_{\delta,t}(x_k | w, \sigma) = \frac{r}{J}$.
10: $\pi = \min_{k=1,\cdots,K} R_{\delta,t}(x_k | w, \sigma)$.
11: $\Delta = \max_{k=1,\cdots,K} ||\delta(x_k) - x_k||_2$.
12: Get $S = s_{(N-m-\lfloor \alpha(N-m+1) \rfloor)}$ from the calibration set $\mathcal{S}_N$.
13: **Output:** If $\Delta < \sigma(\Phi^{-1}(1-S) - \Phi^{-1}(1-\pi))$, the attack is guaranteed to be detected; otherwise, the attack may be detected but without guarantee.

---

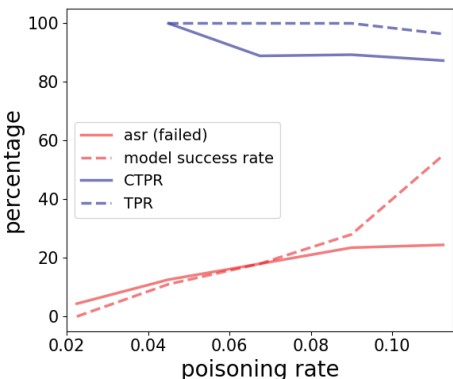

Figure 10: Analysis of the effectiveness of the backdoor attacks with the random perturbation trigger on CIFAR-10. CBD is effective in both detection and certification for a range of poisoning rates.

6.8%, and 9% poisoning rates, respectively. For each poisoning rate, we create 100 attacks and train one model for each attack. The backdoor target class for each attack is randomly selected.

In Fig. 10, we show for each poisoning rate the *model success rate*, which is the proportion of attacks with the (samplewise) attack success rate no less than 90%. We also show the average (samplewise) attack success rate for all the failed attacks for each poisoning rate. With low poisoning rates, the attack may easily fail. But as long as an attack is successful, our CBD is able to detect it with high accuracy, possibly with a guarantee as well.

# I   Difference Between Certified Backdoor Detection and Certified Robustness Against Adversarial Examples

The certification for backdoor detection is fundamentally different from the certification for robustness against adversarial examples from the following five aspects:

- The strength and the stealthiness of an adversarial example are both determined by the adversarial perturbation size. However, for a successful backdoor attack (failed attacks are not supposed to be detected or certified), the attack strength is determined by the robustness of the learned trigger, while the stealthiness is determined by the trigger perturbation magnitude. Thus, for adversarial

examples, a 1-D interval specified by the *certified radius* is derived, while for our CBD, we derive a 2-D *certified region* jointly specified by the trigger robustness and its perturbation magnitude.

- Certification for adversarial examples by randomized smoothing investigates the decision region near a given input. Differently, our certification method for backdoor detection focuses on the model behavior instead, since we are detecting backdoored models. Such model behavior is quantified by our proposed LDP statistic computed on a set of samples from all classes (Definition 3.3).

- Certification for adversarial examples is uni-directional, which provides guarantees on desired model predictions. By contrast, our certification for backdoored model detection is bi-directional. We not only provide guarantees on the detection of backdoored models but also control the false detection rate on benign models. Otherwise, one can easily design a backdoor detector that always triggers an alarm, which provides detection guarantees to all backdoor attacks but is useless in practice.

- For both adversarial examples and backdoor attacks, certified robustness and certified detection have different adversarial constraints – they cover the two ends of the attack strength, respectively. For adversarial examples, existing certifications are all robustness guarantees that cover weak adversarial attacks with small perturbation magnitudes. For backdoor attacks, existing certified defenses provide robustness guarantees on the training-time failure of trigger injection into the model and/or test-time failure of trigger recognition – both failures require the attack to be sufficiently weak. Our CBD, however, focuses on certified backdoor detection, which addresses strong backdoor attacks with robust triggers.

- A strong certified robustness against adversarial examples (and also backdoor attacks) is usually achieved by robust training, which requires the defender to have full access to the training process. However, our certified backdoor detector is deployed post-training, which allows the attacker to have full control of the training process.