# OpenReview forum: "CBD: A Certified Backdoor Detector Based on Local Dominant Probability"
_NeurIPS.cc/2023/Conference — NeurIPS 2023 poster_

### Official Review · Reviewer_XxVd · 2023-06-27

**Soundness:** 3 good
**Presentation:** 3 good
**Contribution:** 3 good
**Rating:** 6
**Confidence:** 3

**Summary:**

The paper presents a certified backdoor detector for backdoor poisoning attacks based on local dominant probability. Certified approaches for backdoor poisoning attacks have been studied recently. These approaches usually have low certified radii on large datasets. This paper goes in another direction by certifying whether the victim model has learned a backdoor attack. The paper can achieve high certified true-positive rates on large datasets because the certified backdoor detection problem is easier than the certified classification under the presence of the backdoored training data.

The key assumption of this paper is that backdoor attacks have high samplewise trigger robustness. Under this assumption, the victim classifier will have a high local dominant probability. The paper uses a small pre-assumed clean validation data to perform conformal prediction and derive bounds using hypothesis testing.

**Strengths:**

1. The paper achieves high empirical and high certified true positive rates on different datasets.
2. It is certifiable.
3. The paper also comes a certified bound of false positive rate under mild assumptions. This certified false positive rate is not provided by other certified backdoor defenses using randomized smoothing.

**Weaknesses:**

1. The paper has the assumption of a small clean validation set.

**Questions:**

Comments:
1. The second last paragraph, "Class imbalance is not the reason for large LDP". This paragraph seems irrelevant. The paper should clarify why it is important to look at class imbalance. Otherwise, the paragraph should be removed.
2. In lines 80-81, these approaches do not allow the attackers to control the full training process, while this paper allows. I suggest the paper emphasizing more on the distinctions of attack models because the distinctions of attack models are more primal than the unavailability of training and test data.
3. In line 97, clarify what is "domain". A set of samples belong to the same label?

Questions:
1. Lines 59-61, where do these numbers come from? Neither Figure 2 nor Table 1.

**Limitations:**

The paper addressed the limitations. For example, the paper has the assumption that backdoor attacks have high sample-wise trigger robustness. At the end of section 5, the paper shows that for the attacks with low sample-wise trigger robustness, one can add noise to the test examples and empirically defense these attacks.

---

> ### Author Rebuttal · Authors · 2023-08-09
>
> # Response to Reviewer XxVd
>
> We sincerely thank you for your time and effort in reviewing our paper and your positive comments. As you mentioned, we investigate certified backdoor defense in a different direction from existing ones by certifying whether the victim model has learned the backdoor. We expect our certified backdoor detection approach, as the first of its kind, to inspire future works that can certifiably detect weaker and less robust backdoor attacks. In the following, we address each of your comments in detail.
>
> _______________________________________
>
> **Q1 :** Assumption of a clean validation set
>
> **A1 :** Thanks for your comment. This is a standard assumption made by most post-training backdoor defenses, though we acknowledge that [1] and [2] perform backdoor detection on simulated data instead. In practice, the defender may collect unlabeled clean data for detection since the backdoored model will provide correct label predictions for these data. We will add more discussion regarding this assumption in our revision.
>
> [1] Chen et al, Deepinspect: A black-box trojan detection and mitigation framework for deep neural networks, 2019.
>
> [2] Dong et al, Black-box detection of backdoor attacks with limited information and data, 2021
>
> **Q2 :** About class imbalance and large LDP
>
> **A2 :** Thanks for pointing this out. As we have claimed in the remarks in **line 138**, the strong robustness of a trigger (which is the basis of a large LDP) is usually achieved by “embedding the trigger in a large variety of samples during training”. However, given that the datasets used by our experiments are generally balanced before the backdoor poisoning, the readers may reasonably hypothesize that the large LDP is also related to the class imbalance caused by poisoning. If this hypothesis is true, classifiers trained on clean but imbalanced data will easily trigger a false alarm. Thus, we presented this paragraph to show that LDP will not be affected by class imbalance in general. In other words, our method indeed detects backdoor attacks, but not the class imbalance in the training data.
>
> We agree that more clarification (e.g. the reasoning above) should be added, and we may move this paragraph to the appendix, depending on the space after revision.
>
> **Q3 :** About the attack model
>
> **A3 :** Thanks for this constructive suggestion! You are right that allowing the attackers to control the full training process is a major distinction and also an advantage of our method compared with the methods mentioned near **lines 80-81**. We will elaborate more on the attack model in our revision.
>
> **Q4 :** About “domain” in line 97
>
> **A4 :** Thanks for pointing this out! The word “domain” refers to the classification domain with a specific sample and label distribution for the model to be inspected. We will explain it carefully in our revision.
>
> **Q5 :** The numbers in lines 59-61
>
> **A5 :** Thanks for raising this issue. The numbers are from **Figure 2** on **page 7**, with the verbal descriptions in **lines 270-272** – sorry that we only included the certified true positive rates but omitted the empirical true positive rates. All these numbers will be added to the caption of **Figure 2** in our revision.
>
> _______________________________________
> Thanks again for your insightful comments, and please let us know any further questions you may have.

---

> > ### Comment · Reviewer_XxVd · 2023-08-12
> > **Response to Authors**
> >
> > Thanks for authors' efforts and detailed reply.
> >
> > For **A2**, thanks for clarifying that. I suggest adding more discussion to talk about the relation between imbalanced data and backdoor poisoning attacks. If the discussion is added, then this experiment totally makes sense.
> >
> > Currently, I'd like to see this paper being accepted. I will read other reviewers questions and comments in the next few days.

---

> > > ### Author Response · Authors · 2023-08-12
> > > **Thank you for your response**
> > >
> > > Thank you for your support and your constructive suggestions! We will add more discussions about the relation between imbalanced data and backdoor poisoning as you suggested. Please let us know if you have any more questions or comments.

---

### Official Review · Reviewer_xfbM · 2023-07-02

**Soundness:** 3 good
**Presentation:** 3 good
**Contribution:** 3 good
**Rating:** 6
**Confidence:** 4

**Summary:**

The paper proposes a certified backdoor detector (CBD) based on a novel, adjustable conformal prediction scheme using a proposed statistic named local dominant probability. The proposed approach claims to be effective against multiple types of backdoor attacks on multiple datasets.

**Strengths:**

The paper is well written, and the theoretical proof is correct to my best knowledge.
I also agree that it is valuable to make certification on backdoor detection.

**Weaknesses:**

1. My concern is that the practical applicability of the certification could be limited. There can be a lot of different backdoor attacks and there a moderate $l_2$ bound only covers a very limited set of attacks. In practice, attackers can still design stealthy triggers (e.g., rotation, backdoor objects, natural corruption, etc.) and such that the trigger can induce a large $l_2$ norm shift. But I understand that $l_p$ bound based certification is already the best that we can do.

2. The overall idea and the applied techniques are quite similar to randomized smoothing for adversarial examples. Though the paper works on backdoor set up, but essentially if $l_2$ bound is assumed, the backdoor problem and the adversarial examples problem seems to just be exactly the same. In this regard, the novelty of this work is a little bit weak and incremental.

**Questions:**

Could you clarify the differences between the certification here and the certification for adversarial examples with randomized smoothing?

**Limitations:**

Limited Practical Applicability.
A little bit incremental considering the randomized smoothing techniques for adversarial examples.

---

> ### Author Rebuttal · Authors · 2023-08-09
>
> # Response to Reviewer xfbM
>
> We sincerely thank you for your time and effort in reviewing our paper. We also thank you for recognizing the importance of our work, the soundness of our theoretical analysis, and the quality of our presentation. In the following, we address your concerns in detail.
> ___
> **Q1 :** Practical applicability of the $\ell_p$ certification
>
> **A1 :** Thank you for your insightful comment. We agree that our certified backdoor detector, as the first of its kind, is limited to a range of backdoor attacks with relatively small trigger perturbation magnitudes in $\ell_2$. The same limitation is also possessed by state-of-the-art backdoor defenses during training that achieves certified robustness against backdoors, such as [1]. Even for certified defenses against adversarial examples, $\ell_p$ certifications are generally not applicable to adversarial patches, while certified defenses against adversarial patches, such as [2], are generally not applicable to $\ell_p$ adversarial perturbations. In this sense, our certification method actually has a decent range of applications, since it covers some patch triggers (BadNet) and blended triggers, though both with relatively small actual perturbation magnitudes in $\ell_2$, as shown by our results (in the parenthesis) in **Table 1**.
>
> We appreciate your understanding of the difficulties in developing certified defenses. Our future works will be devoted to improving the certification bound to cover more trigger types with larger $\ell_2$ norms, such as the rotational trigger, the backdoor object, and the natural corruption, as you mentioned.
>
> [1] Weber et al, RAB: Provable robustness against backdoor attacks, IEEE S&P 2023.
>
> [2] Xiang et al, PatchCleanser: Certifiably robust defense against adversarial patches for any image classifier, USENIX 2022.
>
> **Q2 :** Comparison with the certification for adversarial examples using randomized smoothing
>
> **A2 :** Thanks for your question. Our certification method for backdoored model detection is indeed inspired by the certification for adversarial examples using randomized smoothing due to its general independence of the model architecture. Such independence is important for post-training backdoor detection since a post-training defender has no control of the training process, especially the choice of the model architecture.
>
> However, the certification for backdoor detection is fundamentally different from the certification for adversarial examples from the following *five* aspects:
> * The strength and the stealthiness of an adversarial example are both determined by the adversarial perturbation size. However, for a successful backdoor attack (failed attacks are not supposed to be detected or certified), the attack strength is determined by the robustness of the learned trigger, while the stealthiness is determined by the trigger perturbation magnitude. Accordingly, for adversarial examples, a **1-D interval** specified by the certified radius is derived, while for our CBD, we derive a **2-D certified region** jointly specified by the trigger robustness and its perturbation magnitude (**lines 225-227**).
> * Certification for adversarial examples by randomized smoothing investigates the decision region near a given **input**. Differently, our certification method focuses on the **model** behavior instead, since we are detecting backdoored models. Such model behavior is quantified by our proposed LDP statistic computed on a set of samples from all classes (**Definition 3.3**) -- this is one of the main contributions of our paper, which is clearly different from existing backdoor detectors or certified defenses against adversarial examples.
> * Certification for adversarial examples is **uni-directional**, which provides guarantees on desired model predictions. By contrast, our certification for backdoored model detection is **bi-directional**. We not only provide guarantees on the detection of backdoored models but also control the false detection rate on benign models. Otherwise, one can easily design a backdoor detector that always triggers an alarm, which provides detection guarantees to all backdoor attacks but is useless in practice (**lines 190-192**).
> * For both adversarial examples and backdoor attacks, certified *robustness* and certified *detection* have different adversarial constraints -- they cover the two ends of the attack strength, respectively. For adversarial examples, existing certifications are all robustness guarantees that cover **weak** adversarial attacks with small perturbation magnitudes. For backdoor attacks, existing certified defenses provide robustness guarantees on the training-time failure of trigger injection into the model and/or test-time failure of trigger recognition – both failures require the attack to be sufficiently weak. Our CBD, however, focuses on certified backdoor detection, which addresses **strong** backdoor attacks with robust triggers (**Remark (2), line 189**).
> * A strong certified robustness against adversarial examples (and also backdoor attacks) is usually achieved by robust training, which requires the *defender* to have full access to the **training** process. However, our certified backdoor detector is deployed **post-training**, which allows the *attacker* to have full control of the training process (**lines 78-81**).
>
> Thanks again for your question. We will elaborate more on the differences above in our revision.
> ___
> Thank you again for your insightful comments! Please let us know any follow-up questions you may have.

---

> > ### Comment · Reviewer_xfbM · 2023-08-12
> > **Thanks to Authors**
> >
> > Dear Authors,
> >
> > This is a strong rebuttal. I appreciate your efforts in answering my questions. I will keep my recommendation for acceptance.
> >
> > You did a great job, and hopefully, you can keep pushing forward in this line of work and achieve strong certification for general backdoor attacks in the future, as you mentioned.
> >
> >
> >
> > Thanks,
> >
> > Reviewer xfbM

---

> > > ### Author Response · Authors · 2023-08-12
> > > **Thank you for your response**
> > >
> > > Dear Reviewer xfbM,
> > >
> > > Thank you so much for your recognition of our efforts, your insightful comments, and your strong support! Your words are a great encouragement to us -- we will keep pushing this line of work as you suggested. Please let us know anytime if you have more comments or suggestions.
> > >
> > > Thanks,
> > > Authors

---

### Official Review · Reviewer_fRFm · 2023-07-07

**Soundness:** 4 excellent
**Presentation:** 3 good
**Contribution:** 3 good
**Rating:** 7
**Confidence:** 4

**Summary:**

The authors propose a new method for whether machine learning detecting models have been trained with a backdoor trigger, with the following interesting property: if the trigger has sufficiently small L_2 norm, *AND* the trigger is sufficiently robust to Gaussian noise (i.e., the model will likely return the trigger class even if Gaussian noise has been applied after applying the trigger), then the method is guaranteed to detect that the model has been trained with the trigger.

However, this guarantee alone is only uni-directional: it does not control the false *positive* rate, and  relies on setting a threshold for detection (which in turn determines the meaning of the "sufficiently"'s above). In order to set this threshold, the authors propose a calibration scheme using proxy models trained on a small set of benign samples, and derive p-values using order statistics in order to set the threshold. The authors argue (and empirically verify) that because the proxy models are trained on _less_ data, they will have _more_ variance in the test statistic than a distribution of fully-trained models, and therefore the threshold will be conservative, limiting the false positive rate.

Empirical results on a variety of datasets show improvements over existing detection methods.


**Strengths:**

- I found this to be a very clever method, using randomized-smoothing-like bounds in a novel yet practical way.
- The presentation was easy to follow, and gave an intuitive understanding of what the method is and how its guarantees come about.
- Empirical results are good, comparing to multiple prior works on multiple datasets.

**Weaknesses:**

- The one-sidedness of the certificate makes it less strong of a result than other methods (i.e., for evasion attacks) which are called "certificates." Although the calibration result (Theorem 2) mitigates this somewhat, Theorem 2 still relies on the *empirical* observation that the calibration models have a higher variance in LDP than benign models, and so is not quite air-tight. However, these limitations are well-explained.

- For the methods CBD_sup K-Arm and MNTD, it would be nice to see ROC curves/AUC comparison, rather than just fixing FPR at 5%


Minor issues
- The expectation in the definition of LDP in Equation (1) maybe should be deleted: it seems like the LDP is in practice calculated using only 1 set of samples x_1,...,x_k, and moreover Theorem 1 is defined in terms of the LDP on _those specific samples_. Maybe LDP is just a stochastic quantity?
- The notation in lines 193 vs line 171 is confusing (capital vs lower-case S); is this a typo?
- Section 5.1.3 doesn't seem very useful/is perhaps misleading, and can probably be moved to the appendix: in this test, the calibration and benign samples are drawn from exactly the same distribution, which is not the case in real life (where the calibration set would be trained on less data). Thus the results are just a statistical inevitability, and perhaps not that important to show. (However, in Section 5.3, Figure 4, the empirical fact that the calibration distributions dominate the benign distributions is demonstrated)




**Questions:**

- Could the conformal prediction results be applied to other detection frameworks (K-Arm and MNTD)?


**Limitations:**

Limitations are well-addressed, and this is a purely defensive method, so no clear negative impact.

---

> ### Author Rebuttal · Authors · 2023-08-09
>
> # Response to Reviewer fRFm
>
> Thank you for your time and effort in reviewing our paper, and your positive comments on the novelty and the presentation of our work. We are especially encouraged by your appreciation of our certification approach as reflected in your accurate summary of our work. In the following, we address each of your comments in detail.
> ___
> **Q1 :** Empirical observation related to the assumption used by Theorem 2
>
> **A1 :** Thank you for your comment! As you have pointed out, our probabilistic upper bound on the false positive rate in Theorem 2 does require an assumption about the calibration distribution, which is based on the empirical observation of LDP. Also, as you have mentioned, we have made a lot of efforts in the validation and explanation of this assumption, e.g. in the middle of Figure 4 and in Appendix B.
>
> In this work, we aim to develop a non-trivial certification for backdoor detection with practical usage. And one of the hardest problems for backdoored model detection (even for purely empirical methods) is to set a detection threshold in practice. This is the reason why we endeavored in the analysis of the calibration scheme.
>
>
> **Q2 :** AUC comparison
>
> **A2 :** Thanks for your constructive suggestion. In the table below, we compare our CBD with K-Arm and MNTD in terms of the area under the ROC curves. Our certified CBD achieves AUCs comparable to the uncertified K-Arm on GTSRB and CIFAR-10, and a much larger AUC than K-Arm on SVHN. MNTD achieves a very high AUC on CIFAR-10, but fails for GTSRB and SVHN. In particular, MNTD exhibits an “opposite” behavior on GTSRB, which yields a very small AUC far less than 0.5 – backdoored models tend to have lower scores than benign models in this case.
>
>
> ||GTSRB|SVHN|CIFAR-10|
> |-|-|-|-|
> | MNTD   | 0.11   | 0.49    | 0.97 |
> | K-Arm  | 1.0    | 0.79    | 0.91 |
> | CBD    | 0.98   | 0.97    | 0.89 |
>
> The main purpose for fixing the FPR of K-Arm and MNTD in our paper is to compare them with NC and our CBD both adopting a 0.05 significance level. We agree that the AUC comparison is very informative and we will add the AUC results in the table above (with the ROC curves) to our revision. Thanks again for your suggestion.
>
> **Q3 :** Expectation in the definition of LDP
>
> **A3 :** Yes, LDP is computed on one set of samples $x_1, …, x_k$ in practice and computation on more samples will not change the detection performance much. And you are right that our Theorem 1 considers the actually computed LDPs in the practical detection procedure. So it is better to remove the expectation in Equation (1). Thanks for pointing this out.
>
> **Q4 :** The capital S and the lower-case S
>
> **A4 :** Thanks for pointing this out. In lines 193-194 in our Theorem 2, we considered a random calibration set where the randomness comes from both the model initialization and the data sampling for computing the LDPs. Sorry that we used the same notation in Theorem 2 to represent the calibration set as in line 171 (for the actual calibration set computed in the practical detection procedure). We will change the notation properly in our revision.
>
> **Q5 :** Moving Section 5.1.3 to the appendix
>
> **A5 :** Thanks for your suggestion. We agree that Section 5.1.3 should be moved to the appendix. We will leave the space for the AUC comparison you suggested.
>
> **Q6 :** Applying conformal prediction to K-Arm and MNTD
>
> **A6 :** Thanks for your question. This is a very insightful point since K-Arm and MNTD also need to determine a detection threshold in practice. Theoretically, the conformal prediction results can be applied to K-Arm. But in practice, we will need to run K-Arm on each shadow model to get a score, which will take 382s, 247s, and 240s per model on GTSRB, SVHN, and CIFAR-10, respectively (compared with 8s, 4s, 7s for our CBD).
>
> For MNTD, a feature vector is extracted from each shadow model, which cannot be directly used to distinguish backdoored models from benign ones like our LDP statistic (where a model with a large LDP is likely backdoored). In fact, the authors of MNTD have investigated the possibility of “one-class” training on features extracted from the benign models only, which turned out to be not very effective (the 2nd paragraph on page 2 of their paper). Thus, MNTD and K-Arm may not benefit from our conformal prediction results in practice.
>
> ___
> Thank you again for your time and constructive suggestions. We are happy to answer any follow-up questions you may have.

---

> > ### Comment · Reviewer_fRFm · 2023-08-10
> > **Response to Rebuttal**
> >
> > Thank you for responding to my comments. The AUC numbers are good to see, and the explanation about the applicability of the conformal prediction results to other detection methods was helpful. Overall, I think the method proposed in this paper seems both practically effective and has interesting theoretical properties: I continue to recommend that the paper be Accepted.

---

> > > ### Author Response · Authors · 2023-08-10
> > > **Thank you for your response**
> > >
> > > Thank you for your support and all your constructive suggestions! We will keep improving our paper. Please let us know anytime if you have any follow-up questions. Thank you!

---

### Official Review · Reviewer_jLvZ · 2023-07-18

**Soundness:** 2 fair
**Presentation:** 3 good
**Contribution:** 3 good
**Rating:** 7
**Confidence:** 3

**Summary:**

This paper proposes CBD, a certified detection method for backdoor attacks. It leverages a newly proposed statistic: local dominant probability to capture the target label probability increment in the neighborhood of infected instances. The theoretical analysis provides both detection inference and the condition that detection can be guaranteed. Compared to empirical detectors, CBD also shows comparable or even better detection performance across 4 benchmark datasets and 3 types of attacks.

**Strengths:**

1. The topic is trending and important for the community

2. The paper is overall well-written and easy to follow

3. The proposed method is largely novel and theoretical analysis is sound.

4. The proposed CBD can outperform several state-of-the-art empirical detectors effectiveness wise.

**Weaknesses:**

1. The evaluation regarding the efficiency of CBD is missing

2. Threat model is not consistent with baselines and comparison might be unfair.

**Questions:**

Most of the existing backdoor detectors do not provide theoretical guarantee, which makes them vulnerable to zero-day attacks or trigger types. This paper proposes the first backdoor detection framework with theoretical guarantee. Hence, I believe this paper is valuable. However, I still have several concerns regarding experimental design and details, which might neutralize the performance improvement of the proposed method compared to baselines.

1. In line 259-261, authors mentioned that ‘we train 100 shadow models for each of GTSRB,SVHN, and CIFAR-10, and 50 shadow models for TinyImageNet using the same architectures and  configurations as above – these shadow models are used by our CBD for detection and certification.’ My concern is that training shadow models might significantly increase the detection overhead and make the proposed method time-consuming.  Can you provide the detection overhead of the proposed method under a different number of shallow models and compare them with baseline methods?  It might be impractical to scale the proposed method up to large models(such ImageNet model) if it is slow.

2. In line 235, authors mentioned that ‘We also reserve 5,000 samples from the test set  of GTSRB, SVHN, and CIFAR-10, and 10,000 samples from the test set of TinyImageNet (much smaller than the training size for the models for evaluation) for the defender to train the shadow models. ’  The number of samples needed for CBD is much larger compared to baseline methods, such as K-Arm and NC. Reverse-engineering based detectors usually only require around 10-20 samples to achieve good performance. I am concerned that this sample size discrepancy might cause the comparison unfair. It would be more convincing if authors can show the performance of CBD stays under a smaller number of clean samples.

**Limitations:**

Yes

---

> ### Author Rebuttal · Authors · 2023-08-09
>
> # Response to Reviewer jLvZ
> Thank you for your time and effort in reviewing our paper and your positive comments. In the following, we address each of your concerns in detail.
> ___
> **Q1 :** Time efficiency of CBD
>
> **A1 :** Thank you for your comment. We discuss the time efficiency of CBD from three aspects.
>
> **a. The detection overhead of CBD**
>
> The detection overhead of our CBD is shown in **Appendix E**. For GTSRB, SVHN, CIFAR-10, and TinyImageNet, each shadow model training requires 39s, 45s, 31s, and 2462s, respectively, on a single RTX 2080 Ti card; while the LDP estimation for each model requires merely 8s, 4s, 7s, and 86s on the four datasets respectively. Thus, on GTSRB for example, it will take roughly (39s+8s)xN to get a calibration set of size N.
>
> **b. In practice, existing (uncertified) backdoor detectors, such as K-Arm, also need shadow models to pick a detection threshold.**
>
> Many existing defenses, all uncertified, perform backdoor detection using statistics (e.g. trigger size) derived from the synthesized triggers. Some of them, such as Neural Cleanse (NC), are implemented with an unsupervised anomaly detector to capture any outlier statistics, but are not reliable due to the existence of “natural backdoors” in many practical datasets. Other methods, such as K-Arm, directly set a threshold (e.g.) on the estimated trigger size, which is clearly dependent on the classification domain. Thus, in practice, these methods still need calibration, e.g. using shadow models as we did, to set a detection threshold.
>
> **Comparison**: If a detection threshold is set, K-Arm will need 382s, 247s, and 240s for each model inference on GTSRB, SVHN, and CIFAR-10, respectively, while NC requires 5968s, 1038s, and 816s, respectively – both are much slower than our CBD in per-model inference. Moreover, if K-Arm also trains N shadow models to determine the detection threshold, on GTSRB for example, it will take 382sxN to apply K-Arm to all N shadow models, which is much longer than 8sxN for CBD.
>
> **c. The time efficiency of CBD can be improved, likely without sacrificing the detection accuracy, by using fewer shadow models.**
>
> Theoretically, with fewer shadow models, the expectation of the detection threshold being selected will not change, though its variance will increase. However, we observe that LDPs for backdoored models being detected are generally very large, such that moderate fluctuation in the detection threshold will not significantly affect the performance of CBD.
>
> This is reflected in the results in the table below, where we show the detection accuracy (both empirical and certified (in the parenthesis)) of CBD with $\beta=0.2$ using **10, 25, and 50** shadow models, respectively. These shadow models are randomly sampled from the 100 shadow models we trained for each dataset for the results in our paper. We didn’t compare CBD with the baselines against attacks on TinyImageNet in this work, since training a sufficient number of backdoored models for evaluation on the entire TinyImageNet dataset is very time-consuming.
>
> ||GTSRB||||SVHN||||CIFAR10||||
> |-|-|-|-|-|-|-|-|-|-|-|-|-|
> ||benign|BadNet|CB|Blend|benign|BadNet|CB|Blend|benign|BadNet|CB|Blend|
> |10|0|95(10)|95(85)|95(35)|0|75(45)|100(100)|90(75)|25|85(25)|100(100)| 55(40)|
> |25|0|95(5)|95(85)|95(25)|0|95(45)|100(100)|80(75)|25|80(25)|100(100)|70(40)|
> |50|0|95(15)|95(85)|95(40)|0|95(65)|100(100)|85(75)|25|80(25)|100(100)|60(40)|
>
> **Q2 :** Data efficiency of CBD
>
> **A2 :** Thanks for the comment. Again, reverse-engineering-based detectors, if not equipped with an anomaly detector (which is usually not scalable to a large number of classes), may still need to determine a detection threshold in practice, likely by training a set of shadow models for calibration. In fact, if the calibration set is given, both detection and certification of CBD will only need **one** sample per class (for the estimation of LDP, remarks in line 142), which is much less than the data size required by reverse-engineering-based detectors. Thus, for a fair comparison (e.g.) with K-Arm in practical scenarios, our method actually requires fewer clean samples for detection.
>
> Still, we evaluate CBD with the shadow models trained on fewer samples. In this case, more shadow models will exhibit an abnormally large LDP due to the significantly insufficient training; thus, a larger outlier fraction $\beta$ should be considered. In the table below, we show the detection performance of CBD, with $\beta=0.2$ and $\beta=0.4$, respectively, on GTSRB, SVHN, and CIFAR-10, respectively, with only **100 samples per class** used for training the shadow models. Although we have observed a slight performance drop due to the insufficient training of the shadow models, the detection performance of our CBD is still comparable to the uncertified detectors such as K-Arm.
> ||GTSRB||||SVHN||||CIFAR10||||
> |-|-|-|-|-|-|-|-|-|-|-|-|-|
> ||benign|BadNet|CB|Blend|benign|BadNet|CB|Blend|benign|BadNet|CB|Blend|
> |CBD0.2|0|90(5)|95(85)|90(25)|0|75(45)|100(95)|80(75)|0|35(0)|95(90)|45(40)|
> |CBD0.4|0|95(5)|95(85)|95(35)|0|95(45)|100(100)|85(75)|5|60(10)|100(95)|55(40)|
>
> We also consider an extreme case on GRSRB and CIFAR-10 where only **20 samples per class** are used for training the shadow models. SVHN is not considered here since the shadow models trained on such few samples will have less than 20% test accuracy (i.e. close to random guess). In the table below, we show that CBD can achieve similar detection accuracies and certification results as in Table 1 of the paper if we further increase $\beta$ to 0.8.
> ||GTSRB||||CIFAR10||||
> |-|-|-|-|-|-|-|-|-|
> ||benign|BadNet|CB|Blend|benign|BadNet|CB|Blend|
> |CBD0.2|0|15(0)|80(80)|50(5)|0|45(0)|95(90)|50(40)|
> |CBD0.4|0|30(5)|90(80)|65(5)|5|60(15)|100(95)|55(40)|
> |CBD0.8|0|90(5)|95(85)|90(25)|30|90(40)|100(100)|75(40)|
> ___
> We hope our responses above have well-addressed your concerns. We are happy to answer any further questions you may have.

---

> > ### Comment · Reviewer_jLvZ · 2023-08-21
> > **Thanks for the rebuttal**
> >
> > Thanks authors for the detailed response. Most of my concerns are well addressed. I will raise my score.
> >
> > Good Luck!

---

> > > ### Author Response · Authors · 2023-08-21
> > > **Thank you for your response**
> > >
> > > Thank you for your insightful comments, which are very helpful to our revision! We also thank you for your support, which is very encouraging to us!

---

### Official Review · Reviewer_MHNa · 2023-07-21

**Soundness:** 3 good
**Presentation:** 2 fair
**Contribution:** 3 good
**Rating:** 5
**Confidence:** 3

**Summary:**

This paper proposes a certification method to detect backdoored models. It is based on the assumption that backdoor triggers are robust, meaning that adding random noises onto trigger-injected samples will not change their predictions (to the target label). It also assumes that the neighborhood of a given input (by adding random noises) will be likely predicted as the target as well. This paper hence leverages the output probability difference between clean models and backdoored models when inputs are perturbed with small random noises. It uses a set of shadow models trained on a subset of clean data to facilitate the detection/certification. The evaluation is conducted on four image benchmarks and the results show the proposed method has a reasonable detection/certification performance.

**Strengths:**

1. This paper shows the possibility of conducting certification in the post-training setting, where a model is given without knowing whether it is backdoored or not.

2. It is interesting to leverage the output probability difference between clean and backdoored models when the inputs are perturbed by random noises.

**Weaknesses:**

1. While the perspective of this paper is quite interesting, it is not instantly clear to me why the proposed method can be regarded as a certification method. It is evident that the proposed approach requires training a bunch of shadow models and the certification bound is determined by them. Algorithm Theorem 4.2 provides a theoretical analysis on the usage of benign models, it does not provide a guarantee that those shadow models trained a small subset exhibit the same needed behavior as a benign model trained on the full dataset.

2. Following up the previous point regarding using shadow models, the comparison between SOTA detectors and the proposed approach may not be fair. This paper requires 5000 samples (10000 samples for TinyImageNet) to train shadow models. Such a large number of samples are normally not required for existing detectors. They only needs a few images (less than 10 from each class) to detect backdoored models. For a fair comparison, those baseline techniques should be modified to make full use of the same number of samples for detection.

3. The detection accuracy of NC is unreasonably low on BadNet. According to the original paper and many followup works, NC should not have problem detecting backdoored models by BadNet. The authors should conduct a sanity check to ensure all the baselines are implemented correctly. In addition, the baseline detectors are all from two years ago and dated. There are plenty of recent SOTA detectors, e.g., [23][58]. This paper should at least compare with one or two recent methods.

4. In Section 3.2 at line 110-113, this paper claims that because the backdoor perturbation is usually small, "a significant proportion of samples in the neighborhood of x will also be classified to class t." This is not necessary true. If the backdoor trigger is a yellow square at the top left corner on an image, I don't think adding small random noises on a clean image will lead to the target-class prediction. If this is the case, the model must have very low performance as a small random noise can easily change the prediction.

5. The entire detection/certification procedure is based on the assumption that backdoored models have a higher robustness when the inputs are added with random noises. This paper shows that an existing attack that violates this assumption can be defended by adding random noises during inference. However, an adaptive attacker can intentionally reduce the robustness of backdoor triggers to the same noise level as the proposed approach. In this case, the proposed approach cannot certify the model. Any random noise smaller than that used in the certification also cannot affect the trigger. Note that if using a noise level larger than the certification noise can remove the backdoor effect, there is no need for certification as the problem can be easily solved.

**Questions:**

See above.

---

> ### Author Rebuttal · Authors · 2023-08-09
>
> # Response to Reviewer MHNa
>
> Thank you for your time and effort in reviewing our paper. In the following, we address each of your concerns in detail.
> ___
> **Q1:** Why is the proposed method certified?
>
> **A1:** Thanks for your question. Our certification consists of the following two parts.
>
> For the first part, given any calibration set (or just a detection threshold), backdoor attacks satisfying **Inequality (3)** are guaranteed to be detectable -- based on the existence of such a detection guarantee, our detector is certified.
>
> The second part of our certification is a probability upper bound on the false positive rate to avoid overly aggressive detection. We assume that the LDP distribution for the shadow models has a first-order stochastic dominance over the LDP distribution for the benign models with much more sufficient training (**line 195**). This assumption is empirically validated in the middle of **Figure 4** and is analyzed theoretically in **Appendix B**.
>
> In addition, as the first work to study certified backdoor detection, our certification is non-trivial as shown by the results in **Figure 2 (solid lines)** and **Table 1 (in the parenthesis)**.
>
> **Q2 :** About the fairness in comparison
>
> **A2 :** Thanks for your comment. First, as the best among the three baselines we compared with, K-Arm first performs trigger reverse-engineering and then compares the estimated trigger sizes with a prescribed threshold, which is largely dependent on the classification domain. Thus, in practice, K-Arm also needs to train shadow models to pick a threshold for each domain. In our experiments, we have optimized the thresholds for K-Arm (and also MNTD) on each dataset by maximizing the TPR subject to a 5% FPR (**lines 308-309**). Thus, we actually helped to tune these methods to achieve their best performance. In fact, given a prescribed detection threshold, our CBD requires only **one image per class** for each model inference (**Definition 3.3**) – much fewer than the number of images required by K-Arm.
>
> In addition, following your suggestion, we conducted experiments for CBD with $\beta=0.2$ and $\beta=0.4$, respectively, when the shadow models are trained on only **100 samples per class**. The results in the table below show that CBD can maintain a decent detection and certification performance with only a limited number of data.
>
> ||GTSRB||||SVHN||||CIFAR10||||
> |-|-|-|-|-|-|-|-|-|-|-|-|-|
> ||benign|BadNet|CB|Blend|benign|BadNet|CB|Blend|benign|BadNet|CB|Blend|
> |CBD0.2|0|90(5)|95(85)|90(25)|0|75(45)|100(95)|80(75)|0|35(0)|95(90)|45(40)|
> |CBD0.4|0|95(5)|95(85)|95(35)|0|95(45)|100(100)|85(75)|5|60(10)|100(95)|55(40)|
>
> **Q3 :** Performance of NC
>
> **A3 :** We directly used the NC code from BackdoorBench. As shown in Table 1(a) of [1] and Table 3 of [2], NC achieves around 50% TPR/recall on GTSRB for the BadNet square, which is consistent with our results in Table 1. For CIFAR-10, we used the same model architecture as in [3]. In Table 3 of [3], NC achieves 0.537 AUC on CIFAR-10 for BadNet, which is also consistent with our results in Table 1.
>
> Thank you for pointing out the recent detectors. These methods may perform better than K-Arm, but the performance gain may be small, since K-Arm has already achieved a high detection accuracy. Still, we will include the comparison results in our revision.
>
> [1] Guo et al, TABOR: A highly accurate approach to inspecting and restoring Trojan backdoors in AI systems, 2019.
> [2] reference [64] in our paper
> [3] reference [70] in our paper
>
> **Q4 :** Detection performance against triggers with large perturbation size
>
> **A4 :** Thanks for your comment. Intuitively, whether a BadNet square can be detected using our method depends on the size of the square and the robustness of the learned trigger. Using **Figure 1(b)** for illustration, the smaller the size of the square is, the closer $\delta(x_1)$ is to $x_1$; the stronger the trigger robustness is, the larger the orange area around $\delta(x_1)$ can be. Thus, for any fixed noise std $\sigma$, small square triggers that are robustly learned will likely be detected.
>
> For our results in **Table 1**, we used 2x2 squares on GTSRB and CIFAR-10, and 3x3 squares on SVHN (see **Appendix C4**). Moreover, the noise stds used for GTSRB, SVHN, and CIFAR-10 are 1.15, 0.39, and 1.14, respectively (**line 322**), which are actually not small. This is possibly the reason why our method can still detect some backdoor attacks with a small BadNet square.
>
> **Q5 :** Adaptive attacks and needs for certification
>
> **A5 :** First, in many subfields of trustworthy machine learning, one of the major goals for developing certified defenses is to mitigate potential new adaptive attacks and the follow-up defenses against them (e.g., security cat and mouse game). For the robustness against adversarial examples, for instance, an attacker can always find an adversarial perturbation that effectively fools a model by increasing the perturbation size. However, certified defenses guarantee a desired output when the adversarial perturbation is within some certified radius. In other words, if the perturbation budget is less than the certified radius, an adaptive attacker is guaranteed to fail.
>
> For backdoor attacks, existing certified defenses we cited in **line 80** provide guarantees on the training-time failure of trigger injection into the model and/or test-time failure of trigger recognition – both failures require the attack to be sufficiently weak and satisfy certain adversarial constraints (i.e. with low trigger robustness on defenseless models). Our CBD, however, focuses on backdoor attacks with strong trigger robustness, which is complementary to existing certified backdoor defenses (**Remark (2), line 189**). We expect this certification gap to be closed by future works considering diverse adversarial constraints.
> ___
> Thanks again for your insightful comments and suggestions. Please let us know any further questions you may have.

---

> > ### Comment · Reviewer_MHNa · 2023-08-15
> > **Thanks for the rebuttal**
> >
> > Thanks for providing the response.
> >
> > According to the response to Q4, the submission actually uses a quite high noise level to perturb the input. This was not clearly stated early in the paper such as at line 110-113, which causes confusing. It would be better if this could be noted early in the paper to help easy understanding.
> >
> > The concerns are sufficiently addressed. I will raise my score.

---

> > > ### Author Response · Authors · 2023-08-15
> > > **Thank you for your response**
> > >
> > > Thank you for your insightful comments and constructive suggestions, and for raising the score. We will provide more details about the noise level in the early part of the paper, as you suggested. Please let us know if you have more comments or questions. Thanks again for your time!

---

### Author Rebuttal · Authors · 2023-08-09

# Summary

We sincerely thank the reviewers for their constructive feedback and suggestions. We are glad that the reviewers found our paper novel and effective. Our responses are summarized as follows:

* Experiments added as suggested by the reviewers:

1. We showed the performance of our CBD with much fewer shadow models.
2. We showed the performance of our CBD with much fewer samples used for training the shadow models.
3. We compared our CBD with K-Arm and MNTD in terms of the area under the ROC curves.

* Clarification and revision following the comments of reviewers:

1. We discussed the actual detection overhead of our CBD (and the other methods we compared with) for practical detection procedures.
2. We discussed the detection and certification capability of our method in practice against backdoor attacks with large perturbation magnitude in $\ell_2$.
3. We discussed the actual data size required by our CBD (and the other methods we compared with) for practical detection procedures.
4. We showed that the results of Neural Cleanse (NC) in our paper are consistent with prior works.
5. We explained why our method is certified and made it more clear.
6. We justified the importance of certification in backdoor detection.
7. We explained the assumption required by Theorem 4.2.
8. We discussed the possibility of applying our conformal prediction results to other detection frameworks.
9. We listed five fundamental differences between the certification in this paper and the certification for adversarial examples using randomized smoothing.
10. We explained the assumption of the clean dataset for detection.
11. We explained why it is meaningful to investigate the class-imbalance scenarios.
12. We will move Section 5.1.3 to the appendix as suggested by reviewer fRFm.
13. We will make an editorial pass to address the notation issues pointed out by reviewer fRFm.
14. We will add more discussion regarding the attack model as suggested by reviewer XxVd.
15. We will elaborate on the concepts and the numbers pointed out by reviewer XxVd.

We hope all of your concerns have been well-addressed in our responses. We are more than willing to address additional questions and conduct further experiments should the reviewers deem it necessary.

---

### Decision · Program_Chairs · 2023-09-21

**Decision:**

Accept (poster)

**Comment:**

This paper proposes a certified backdoor detector which not only predicts whether a model has been attacked, but also provides conditions of guaranteed detection, and an upperbound on false positive rates. The method and the theoretical result, although requiring a validation set, are considered novel and sound by reviewers. Concerns regarding evaluation, baselines, and experimental settings were raised, but were well addressed during rebuttal.